# LABEL-AGNOSTIC ATTRIBUTION FOR INTERPRETABILITY

## ABSTRACT

The importance of attribution algorithms in the AI field lies in enhancing model transparency, diagnosing and improving models, ensuring fairness, and increasing user understanding. Gradient-based attribution methods have become the most critical because of their high computational efficiency, continuity, wide applicability, and flexibility. However, current gradient-based attribution algorithms require the introduction of additional class information to interpret model decisions, which can lead to issues of information ignorance and extra information. Information ignorance can obscure important features relevant to the current model decision, while extra information introduces the incorrect identification of irrelevant features as significant. To address these issues, we propose the Label-Agnostic Attribution for Interpretability (LAAI) algorithm, which analyzes model decisions without the need for specified class information. Additionally, to more rigorously assess the potential of current attribution algorithms, we introduce a variety of new evaluation metrics, combined with the traditional Insertion & Deletion Scores, to comprehensively assess the performance of our algorithm. To continuously advance research in the field of explainable AI (XAI), our algorithm is open-sourced at `https://anonymous.4open.science/r/LFAI-336C`.

## 1 INTRODUCTION

As deep learning advances, performance on tasks such as image recognition (Xu et al., 2023; Liu et al., 2023) has reached unprecedented levels, driving transformative applications in healthcare (Suganyadevi et al., 2022), autonomous driving (Grigorescu et al., 2020), and managerial decision-making (Shrestha et al., 2021). This growing reliance heightens the demand for transparent decisions: without explainability, users struggle to trust results or assign responsibility when failures occur.

Therefore, the research and development of Explainable AI (XAI) are of paramount importance. There has been extensive research in the XAI domain, with early interpretability methods such as Grad-CAM (Selvaraju et al., 2017) and LIME (Ribeiro et al., 2016) using heatmaps and local linear models to explain the decisions of Deep Neural Networks (DNNs). However, these methods have limitations in providing fine-grained and one-to-one explanations for each input feature. Consequently, researchers have proposed more detailed attribution methods, with Integrated Gradients (IG) (Sundararajan et al., 2017) being one of the most significant. IG addresses the shortcomings of earlier methods and introduces axioms for attribution, providing a consistent and fair framework for feature importance. As research progressed, new adversarial example-based attribution methods were proposed, such as Adversarial Gradient Integration (AGI) (Pan et al., 2021), MFABA (Zhu et al., 2024), and AttEXplore (Zhu et al., 2023).

We note that existing attribution methods typically select specific class outputs or cross-entropy as the loss function and use backpropagation to obtain gradients concerning input samples to guide the attribution algorithm. We have identified two phenomena that cause attribution bias due to such gradient selection: **information ignorance and extra information.** Information ignorance refers to the omission of important features from classes not directly related to the model's final decision, while extra information involves the incorrect identification of irrelevant features as significant. The former leads to interpretability methods overlooking many features crucial to the model's current decision and failing to explain low-confidence situations (applicable under any less-than-100% con-

fidence conditions). And the latter results in feature leakage (Shah et al., 2021), where features not contributing to the model's decision are incorrectly identified as important.

Instead of requiring label information as a hard prerequisite, we explore a complementary *label-agnostic* perspective on attribution. Models are indeed trained with labels, and label-conditioned explanations (e.g., "why is this image a dog?") remain important and widely used. However, many applications also demand answers to a different question: *which features drive the model's current predictive distribution?* without committing to any particular class. From this perspective, the model's output is fully determined by the input features, and the role of attribution is to characterize how these features shape the entire distribution rather than to justify a pre-selected label.

When explanations are strictly tied to a single label—either the ground-truth or the top-1 prediction—the gradient $\nabla_x L(f(x), y)$ is forced to answer "why $y$?", which can systematically under-represent features that primarily affect competing classes (i.e., *information ignorance*) and over-emphasize features that merely support the hypothesis "this input is $y$" even when the predictive distribution is ambiguous (i.e., *extra information*). *To specify certain information always means to ignore what is not specified.* It is also worth noting that even using the highest-confidence class as a form of supervision still involves label information, as this choice effectively inflates the confidence of the selected class and suppresses alternatives. In this work, we explicitly study label-agnostic attributions that operate directly on the predictive distribution and are thus designed to mitigate these two failure modes.

To address these phenomena, we propose a **Label-Agnostic Attribution for Interpretability (LAAI)** algorithm. In LAAI, we redefine the form of accumulated gradients, eliminating the need to incorporate class information into the gradients. Additionally, we conduct rigorous mathematical derivations to ensure the validity of the LAAI algorithm and its adherence to attribution axioms (Sundararajan et al., 2017).

Beyond the extra information phenomenon introduced by attribution algorithms, it can also occur during the evaluation of attribution algorithms. This happens because neural networks cannot distinguish between feature removal behavior and the information introduced by adding a baseline. For image tasks, removing features and replacing their values with zero might be interpreted by the neural network as introducing black baseline information, which is unfair for tasks where black is a key feature (e.g., black-and-white cat classification). Thus, we propose the fair insertion and fair deletion metrics to avoid bias during the evaluation process. Additionally, to assess the impact of confusion, we introduce KL insertion and KL deletion metrics. We summarize our contributions as follows:

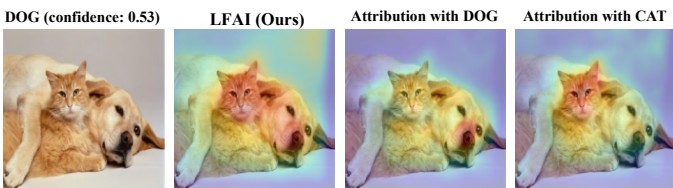

DOG (confidence: 0.53)    LFAI (Ours)    Attribution with DOG    Attribution with CAT

Figure 1: Illustration of the **Information Ignorance** phenomenon. Attribution results using the AttEXplore (Zhu et al., 2023) algorithm on an image containing both a cat and a dog. The model's classification confidence is 0.53, indicating that the model considers both the cat and the dog when making its decision. However, traditional attribution methods only focus on the predefined label, in this case, the dog, and ignore the cat's features, leading to the Information Ignorance phenomenon. In contrast, our method, which does not rely on predefined labels, is able to attribute the features of both the cat and the dog, avoiding the Information Ignorance issue.

**1)** We systematically pinpoint two phenomena that cause bias in current gradient-based attribution algorithms: **information ignorance** and **extra information**, which significantly undermine the reliability of interpretability algorithms; **2)** We propose two new evaluation metrics based on Insertion and Deletion Scores to address the extra information phenomenon in the evaluation of attribution algorithms. To ensure the comprehensiveness of our experimental analysis, we also incorporate the traditional Insertion and Deletion Scores in our evaluation; **3)** Based on the first contribution listed above, we design a novel gradient accumulation method and propose the LAAI algorithm, supported

by rigorous mathematical derivations to ensure its stability; **4)** We open-source our experimental code to facilitate subsequent research and replication of experiments.

## 2 RELATED WORK

The development of interpretability methods for deep neural networks (DNNs) has progressed through three stages: local approximation, gradient-based attribution, and adversarial example-based attribution. Local approximation methods, like LIME (Ribeiro et al., 2016), build simple surrogate models near the input to approximate model behavior. While locally interpretable, such methods are inefficient and rely on potentially flawed assumptions. Layer-wise relevance propagation (LRP)(Bach et al., 2015) and DeepLIFT(Shrikumar et al., 2017) assign importance scores based on reference comparisons, but their sensitivity to baselines and lack of implementation invariance (Sundararajan et al., 2017) lead to inconsistent results. These limitations are further discussed in Zhu et al. (2023; 2024).

Gradient-based attribution methods utilize gradients to trace model predictions to input features. Early methods like Saliency Map (SM)(Simonyan et al., 2013) suffer from gradient saturation. Grad-CAM(Selvaraju et al., 2017) and Score-CAM (Wang et al., 2020) use intermediate feature maps but lack fine-grained precision. IG (Sundararajan et al., 2017) addresses some of these issues by integrating gradients along input paths, though its computational cost is high, prompting fast variants like Fast IG (FIG)(Hesse et al., 2021). Other refinements include EG(Erion et al., 2021), SG (Smilkov et al., 2017), and GIG (Kapishnikov et al., 2021), each introducing stability or selectivity at the expense of added assumptions or potential biases.

Adversarial example-based methods improve attribution by exploring model decision boundaries without requiring manual baselines. These include AGI (Pan et al., 2021), BIG (Wang et al., 2021), AttEXplore (Zhu et al., 2023), and MFABA (Zhu et al., 2024), which respectively introduce nonlinear path integration, boundary-based baselines, parameter exploration, and second-order approximations to enhance fidelity and efficiency. However, such methods may introduce out-of-distribution (OOD) artifacts during adversarial search, resulting in attribution bias.

Importantly, most gradient-based methods rely on cross-entropy or the top-class output as the attribution loss, which we show in Section 3.2 to cause two forms of bias: information ignorance and extra information. These phenomena significantly distort attribution results and motivate our label-agnostic alternative. While Crabbé & van der Schaar (2022) introduce label-agnostic explanations for unsupervised representation learning, their approach does not address pixel-level attributions in supervised classifiers; our LAAI method directly fills this gap by providing label-agnostic feature attributions with new fidelity metrics for supervised prediction tasks.

## 3 METHOD

### 3.1 PROBLEM DEFINITION

Given the neural network parameters $w \in \mathbb{R}^m$ and the sample to be attributed $x \in \mathbb{R}^n$, our goal is to use attribution methods to obtain the attribution result $A(x) \in \mathbb{R}^n$, where $A_i(x)$ represents the importance of the $i$-th feature dimension, $n$ represents the number of dimensions. The larger the attribution result, the more important that dimension is for the model's decision, which means that removing the feature would have a greater impact on the model's decision for the current sample. We use $f_j(x) \in \mathbb{R}$ to represent the model's output for the $j$-th class, and $P_j(x)$ to denote the probability of the $j$-th class after applying the softmax function.

### 3.2 INFORMATION IGNORANCE AND EXTRA INFORMATION

In current gradient-based attribution methods, $\frac{\partial L(f(x),y)}{\partial x}$ is typically chosen as the gradient, where $y$ is the pre-specified ground-truth class or $y = \arg\max_j f_j(x)$, and $L$ is usually the negative of the output value of the class with the maximum output or the cross-entropy loss function. This introduces "Extra Information," which is detrimental to the attribution algorithm. Additionally, since

the loss function $L(f(x), y)$ focuses on the class information of $y$, this leads to the phenomenon of Information Ignorance we will introduce below.

**Information Ignorance** *(Informal)*: Information Ignorance refers to the tendency of attribution methods to overlook the feature information of classes other than the target class. For example, when the target class is the dog, attribution methods focus solely on the dog's features, ignoring the cat's features, and vice versa. However, in reality, the model uses features from both classes during decision-making. As shown in Figure 1, when the confidence for the dog class is 0.53, the model considers both the dog's and the cat's features. This suggests that, despite the focus on the target class, the model integrates information from both classes.

Formally, for a fixed classifier $f$, input $x \in \mathbb{R}^d$, and an attribution method $A$ that produces scores $a_i(x)$ for each feature $i \in \{1, \dots, d\}$, we first fix an (oracle) relevant set $\Phi(x) \subseteq \{1, \dots, d\}$ and thresholds $k \geq 1$ and $\tau > 0$. We then define

$$S_{\text{II}}(x) := \{ i \in \Phi(x) \mid a_i(x) < \tau \}. \tag{1}$$

We say that $A$ exhibits Information Ignorance at level $(k, \tau)$ on $x$ if $|S_{\text{II}}(x)| \geq k$, i.e., at least $k$ truly relevant features receive attribution scores below the threshold $\tau$.

**Extra Information** *(Informal)*: Extra Information refers to the inclusion of features from non-target classes that are not relevant for the decision process, which can mislead the attribution algorithm. In fact, the model initially inputs all features. Although the relative importance of these features varies, our aim is to discern which areas are crucial to the model's decisions and which are not. If certain unnecessary features are retained, they can blur the boundary between important and unimportant features. Formally, we define

$$S_{\text{EI}}(x) := \{ i \notin \Phi(x) \mid a_i(x) \geq \tau \}, \tag{2}$$

and say that $A$ exhibits Extra Information at level $(k, \tau)$ on $x$ if $|S_{\text{EI}}(x)| \geq k$, i.e., at least $k$ irrelevant features are assigned spuriously large attribution scores above $\tau$.

Here, $\Phi(x)$ is an oracle set of truly decision-relevant features used only to conceptually define Information Ignorance and Extra Information. In practice, $\Phi(x)$ can be approximated via object masks, synthetic ground truth, or counterfactual evidence, but our evaluation metrics (INS/DEL, F-INS/F-DEL, KL-INS/KL-DEL) are computed directly from model predictions and do not assume access to $\Phi(x)$. Throughout this paper, $(k, \tau)$ are treated as fixed global hyperparameters (e.g., on normalized attribution scales) and are not tuned per sample or per method, so that the above existence conditions are not made trivial by adjusting $\tau$ or $k$.

We first gain an intuitive understanding of the Information Ignorance phenomenon from the definition of the gradient. By performing a first-order Taylor expansion of the loss function as in $L(f(x + \Delta x), y) \approx L(f(x), y) + \Delta x^\top \cdot \frac{\partial L(f(x), y)}{\partial x} + \mathcal{O}$, where $\mathcal{O}$ represents higher-order infinitesimals (which are ignored in the first-order analysis), we can observe the sensitivity of different dimensions in $x$ to the loss function $L(f(x), y)$. Therefore, SM (Simonyan et al., 2013) directly uses $\frac{\partial L(f(x), y)}{\partial x}$ as the interpretability result. Accumulating $\frac{\partial L(f(x), y)}{\partial x}$ during the change process of the sample $x$ reveals the overall performance of sensitivity, which is the idea behind methods such as AGI (Pan et al., 2021), BIG (Wang et al., 2021), MFABA (Zhu et al., 2024), and AttEXplore (Zhu et al., 2023). However, it is important to note that $\frac{\partial L(f(x), y)}{\partial x}$ only focuses on the loss function $L(f(x), y)$, which in turn only focuses on the class $y$. This means that attribution methods using $\frac{\partial L(f(x), y)}{\partial x}$ will ignore information from classes other than the specified class $y$. When the confidence of $y$ is low, additional features of the class may be introduced, resulting in Extra Information.

**Examples of Information Ignorance**: As shown in Figure 1, the AttEXplore (Zhu et al., 2023) algorithm only provides information about the dog class, although the model also considers the cat area in its decision-making process. The Information Ignorance phenomenon worsens when the confidence of the current class is low. This phenomenon aligns with the formulation in Equation 1, which highlights the inability of the model to account for competing features when the model's confidence is low. However, in reality, the low confidence in the model's output itself also needs to be explained. In other words, **current gradient-based attribution methods cannot explain why the model has low confidence in its decision** (e.g., the presence of a cat explains why the dog class confidence is only 0.53, which could be due to the model focusing on both dog and cat features). The LAAI algorithm, introduced below, avoids Information Ignorance by not using class information as

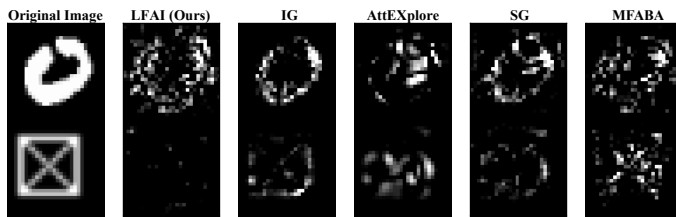

Figure 2: Illustration of the **Extra Information** phenomenon. As shown in the image, ⊠ represents extra information that does not belong to any class feature. The model does not attend to this region, yet other attribution methods, apart from ours, display that the model focuses on the ⊠ feature. This demonstrates that our method can effectively avoid the Extra Information phenomenon.

guidance. More experimental results on LAAI for low-confidence data can be found in Section 4.4 and Appendix F.5.

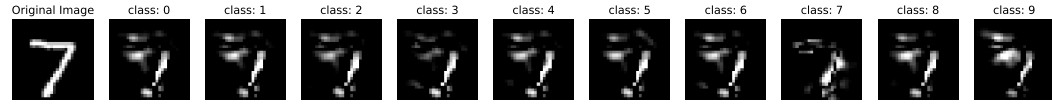

Figure 3: Attribution results for digit recognition using AttEXplore. No significant class differences observed when changing the accumulated gradient class.

Next, we analyze the causes of the Extra Information phenomenon. Previous attribution algorithms, in the process of defining the loss function class, often utilize the output of the class with the maximum activation, inadvertently introducing Extra Information. Intuitively, this approach imposes the assumption that the input sample belongs to the current class onto the interpretability method, which may lead to biased attribution results (Shah et al., 2021).

**Examples of Extra Information**: As illustrated in Figure 2, attribution methods can highlight irrelevant regions that do not belong to any class in the original image. The square patch at the bottom is unrelated to any class. However, when attribution is forced toward class0—even under low-confidence conditions—some methods still ascribe importance to this area, introducing attribution bias. This phenomenon is consistent with Equation 2. Although LAAI and MFABA faithfully capture the stroke and contour features of the digit "0" that the model actually uses (upper part of the image), other methods (e.g., IG, AttEXplore, SG) look more intuitive to humans yet do not reflect the model's true reliance. Crucially, all methods except LAAI—including MFABA—still exhibit *Extra Information* in the irrelevant square region.

An example of this phenomenon is the feature leakage problem discussed in (Shah et al., 2021), which directly manifests as *Extra Information*. In handwritten digit recognition (Figure 2), the background grid pattern is task-irrelevant. Yet, under low-confidence settings, several interpretability methods mistakenly treat these grids as belonging to one of the ten classes, thereby injecting redundant information and biasing the attribution. In other words, the inclusion of irrelevant features leads to information "leakage," where the interpretability process is misled by extraneous data.

### 3.3 DOES LABEL GUIDANCE TRULY GUIDE?

There is a possibility that specifying the corresponding class can yield results that indicate the importance of features for that class. Therefore, we further explore the notion that specifying the target class might assist in identifying important features for that class. However, as demonstrated in Figure 3, when using AttEXplore and modifying the accumulated gradient class for digits 0-9, no significant class-specific differences are observed for digit 7. Similarly, in Figure 4, applying attribution with the cat class on a dog image highlights features that correspond to the dog. This may indicate a model training issue, where features of the dog are misclassified as belonging to the cat class. Masking the core feature area of the dog shifts the attribution focus to the cat, suggesting that the model is able to respond to the correct features associated with the cat class. This observation leads us to hypothesize that, during the accumulated gradient process, changes in lower-

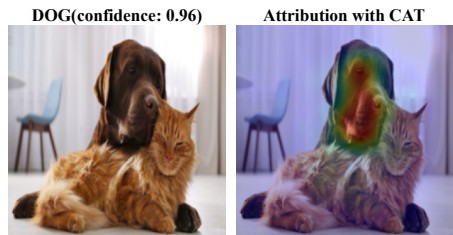

Figure 4: Attribution results using the cat class on a dog image. The middle image shows the attribution result where the model mistakenly highlights dog-related features even when guided by the cat label, suggesting confusion in feature attribution. The right image shows the attribution after masking the core dog region, which shifts attention toward the cat, indicating that the model does have class-specific responses when dominant misleading features are suppressed.

confidence classes are more limited (Zhu et al., 2023), making the class specification process not always effective in guiding the attribution method.

**Remark 1.** *Specifying class information in the gradient calculation during attribution can lead to information loss and the phenomenon of Extra Information.*

### 3.4 Two Phenomena in Attribution Evaluation

Both the Information Ignorance and Extra Information phenomena exist in the evaluation of attribution algorithms. An example of Information Ignorance can be observed in Figure 1. If the evaluation is conducted using the dog class, the importance of the cat features is ignored in the process. However, the model also attends to the cat's features, and the low confidence in the dog class may be a result of the model focusing on the cat's features, thus reducing the confidence in the dog class. Therefore, during the evaluation of explainability methods, it is important to also assess the causes of such low-confidence outcomes.

Additionally, the core idea behind evaluating attribution algorithms involves inserting information from the original image into an all-black pixel image based on the attribution score from high to low, assessing how quickly the current model decision can be recovered, which corresponds to the Insertion score. Conversely, the Deletion score replaces the original image pixels with black pixels from high to low based on the attribution score, evaluating how quickly the current model decision can be destroyed. However, this process inherently assumes that a black image represents "no information." In reality, **the model cannot distinguish between feature removal and features represented by black pixels. This leads to the black pixel information in the baseline becoming additional irrelevant information.** For example, in the task of distinguishing between black and white cats, the color of the cat's fur is an important feature. Zeroing out the features during this process makes a white cat more likely to be seen as a black cat, introducing black cat features rather than removing features.

To replace the zero-out operation with a more reasonable feature removal behavior, we designed the **Confusion Feature Algorithm (CFA)**, the implementation details are presented in Algorithm 1 in Appendix C.

$$x^* = \arg \max_x \mathcal{H}(x) = -\sum_{j=1}^{C} P_j(x) \log P_j(x) \tag{3}$$
$$\text{s.t.} \quad x_1 = x_2 = x_3 = \cdots = x_n$$

The core idea of CFA is to find the pixel value that maximizes the entropy of the model output distribution (the higher the entropy, the greater the uncertainty of the model decision) and ensure that the entire input consists of the same pixel value, aligning with human intuition of feature removal by replacing large areas with the same pixel, as shown in Figure 5. This approach makes the explanation process more consistent with human understanding. Because a single pixel value is easier for humans to comprehend, interpretability methods should aim for results that closely approximate human intuition.

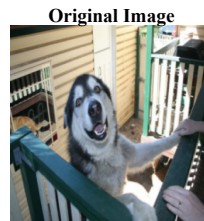 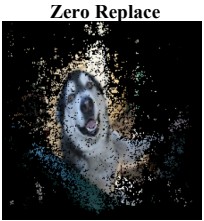 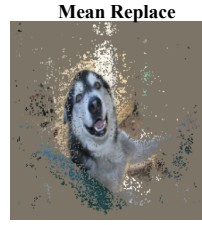

Figure 5: Comparison of Different Feature Replacement Methods

Table 1: Evaluation of various interpretability methods via INS and DEL metrics. ↑ indicates that higher values in the column correspond to better interpretability performance, while ↓ indicates that lower values correspond to better interpretability performance. The * symbol denotes the primary reference metric for comparison. The table is divided into three confidence-based categories: Low Confidence (<70%), High Confidence (≥70%).

| | Inception-v3 | | | | | | ResNet-50 | | | | | | VGG16 | | | | | |
| | Low Confidence (<70%) | | | High Confidence (≥70%) | | | Low Confidence (<70%) | | | High Confidence (≥70%) | | | Low Confidence (<70%) | | | High Confidence (≥70%) | | |
| Method | INS (↑) | DEL (↓) | GAP* (↑) | INS (↑) | DEL (↓) | GAP* (↑) | INS (↑) | DEL (↓) | GAP* (↑) | INS (↑) | DEL (↓) | GAP* (↑) | INS (↑) | DEL (↓) | GAP* (↑) | INS (↑) | DEL (↓) | GAP* (↑) |
|---|---|---|---|---|---|---|---|---|---|---|---|---|---|---|---|---|---|---|
| SM | 0.0226 | 0.0349 | -0.0123 | 0.0889 | 0.047 | 0.0419 | 0.0283 | 0.015 | 0.0133 | 0.0658 | 0.0369 | 0.0289 | 0.0216 | 0.0137 | 0.0079 | 0.054 | 0.0228 | 0.0312 |
| IG | 0.0228 | 0.0283 | -0.0055 | 0.1009 | 0.0312 | 0.0697 | 0.0228 | 0.0108 | 0.012 | 0.0519 | 0.0254 | 0.0265 | 0.0164 | 0.0103 | 0.0061 | 0.0392 | 0.0176 | 0.0216 |
| FIG | 0.028 | 0.019 | 0.009 | 0.0467 | 0.0744 | -0.0277 | 0.0119 | 0.0211 | -0.0092 | 0.0327 | 0.0466 | -0.0139 | 0.0119 | 0.0157 | -0.0038 | 0.0218 | 0.0356 | -0.0138 |
| BIG | 0.0904 | 0.0328 | 0.0576 | 0.1843 | 0.0844 | 0.0999 | 0.047 | 0.0294 | 0.0176 | 0.118 | 0.0693 | 0.0487 | 0.0331 | 0.021 | 0.0121 | 0.088 | 0.0501 | 0.0379 |
| MFABA | 0.0975 | 0.0331 | 0.0644 | 0.2799 | 0.0859 | 0.194 | 0.0504 | 0.0305 | 0.0199 | 0.1401 | 0.078 | 0.0621 | 0.0369 | 0.022 | 0.0149 | 0.1133 | 0.0529 | 0.0604 |
| AttEXplore | 0.1324 | 0.0321 | 0.1003 | 0.3757 | 0.0739 | 0.3018 | 0.0992 | 0.0226 | 0.0766 | 0.2745 | 0.053 | 0.2215 | 0.0807 | 0.0205 | 0.0602 | 0.2444 | 0.0468 | 0.1976 |
| GIG | 0.0242 | 0.0234 | 0.0008 | 0.0992 | 0.0287 | 0.0705 | 0.0225 | 0.0103 | 0.0122 | 0.0517 | 0.0183 | 0.0334 | 0.0187 | 0.0094 | 0.0093 | 0.0409 | 0.0128 | 0.0281 |
| EG | 0.1697 | 0.1848 | -0.0151 | 0.398 | 0.4521 | -0.0541 | 0.1162 | 0.1243 | -0.0081 | 0.2988 | 0.3465 | -0.0477 | 0.0924 | 0.0772 | 0.0152 | 0.2217 | 0.1972 | 0.0245 |
| DeepLIFT | 0.0269 | 0.0209 | 0.006 | 0.0811 | 0.0573 | 0.0238 | 0.0189 | 0.0121 | 0.0068 | 0.0445 | 0.0333 | 0.0112 | 0.0165 | 0.0105 | 0.006 | 0.0355 | 0.0198 | 0.0157 |
| SG | 0.0423 | 0.0299 | 0.0124 | 0.1957 | 0.0251 | 0.1706 | 0.0663 | 0.0105 | 0.0558 | 0.1317 | 0.0175 | 0.1142 | 0.0545 | 0.0098 | 0.0447 | 0.1296 | 0.0134 | 0.1162 |
| AGI | 0.1104 | 0.029 | 0.0814 | 0.374 | 0.084 | 0.29 | 0.0905 | 0.0262 | 0.0643 | 0.3684 | 0.0654 | 0.303 | 0.0537 | 0.0189 | 0.0348 | 0.2935 | 0.0475 | 0.246 |
| LAAI (Ours) | 0.2119 | 0.0385 | **0.1734** | 0.5131 | 0.1056 | **0.4075** | 0.1436 | 0.031 | **0.1126** | 0.3859 | 0.0678 | **0.3181** | 0.1026 | 0.0219 | **0.0807** | 0.3118 | 0.0492 | **0.2626** |

Since finding the optimal solution for Equation 3 is difficult, we use gradient-based iterations to approximate it. Let $\{x^t\}_{t=0}^T$ denote the intermediate iterates, where all spatial locations share the same value $x^t$ and the features input into the network are obtained by repeating a single pixel. The proof is provided in Appendix A.

$$x^t = x^{t-1} - \alpha \cdot \text{sign}\left(\frac{\partial \sum_{j=1}^{C} \log P_j(r(x^{t-1}))}{\partial x}\right) \tag{4}$$

Here, $x^0 \sim U(0,1)$, $U$ represents a uniform distribution (this operation maps pixel values to 0–1 through normalization). $x^0 \in \mathbb{R}^3$ for image tasks, representing the RGB values of one pixel. We denote the update at step $t$ by $\Delta x^t = x^t - x^{t-1}$ and constrain it within a small $\ell_\infty$-ball, i.e., $\|\Delta x^t\|_\infty \leq \varepsilon$, where $\varepsilon > 0$ is a step-size bound and $\alpha > 0$ is the learning rate; $T$ is the total number of iterations. The function $r(\cdot)$ is a repeat operator that tiles the single-pixel value $x^t$ to match the input dimensions of the neural network. To avoid local optima, we sample multiple times and select the $x$ that maximizes $\mathcal{H}(x)$ as the final choice. Replacing the all-black image in the attribution evaluation process with the learned $x$ avoids the Extra Information phenomenon mentioned above. We name the replaced algorithms as Fair Insertion and Fair Deletion metrics. Additionally, to evaluate the model's uncertainty in the attribution process, we propose KL Insertion and KL Deletion metrics, assessing the change in model decision confusion during the attribution insertion process. Traditional Insertion/Deletion (INS/DEL) requires specifying a class for evaluation, bringing potential bias. Our KL-INS and KL-DEL remove the need to designate a specific class in either the evaluation or the attribution. Additionally, our LAAI method has complexity comparable to AGI (Pan et al., 2021). Each time we compute Equation 4, we perform both a forward pass and a backward pass. This means that during the update process, there will be $m \times T$ forward and backward propagations, where $m$ is the number of samples and $T$ is the number of iterations.

Table 2: Evaluation of various interpretability methods via KL-INS and KL-DEL metrics.

| | Inception-v3 | | | | | | ResNet-50 | | | | | | VGG16 | | | | | |
|---|---|---|---|---|---|---|---|---|---|---|---|---|---|---|---|---|---|---|
| | Low Confidence (<70%) | | | High Confidence (≥70%) | | | Low Confidence (<70%) | | | High Confidence (≥70%) | | | Low Confidence (<70%) | | | High Confidence (≥70%) | | |
| Method | KL-INS (↑) | KL-DEL (↓) | GAP* (↑) | KL-INS (↑) | KL-DEL (↓) | GAP* (↑) | KL-INS (↑) | KL-DEL (↓) | GAP* (↑) | KL-INS (↑) | KL-DEL (↓) | GAP* (↑) | KL-INS (↑) | KL-DEL (↓) | GAP* (↑) | KL-INS (↑) | KL-DEL (↓) | GAP* (↑) |
| SM | 4.0006 | 4.1461 | -0.1455 | 4.2733 | 4.2992 | -0.0259 | 5.7474 | 5.7749 | -0.0275 | 5.8957 | 6.0216 | -0.1259 | 4.0809 | 4.0889 | -0.008 | 4.1862 | 4.3423 | -0.1561 |
| IG | 4.3213 | 4.478 | -0.1567 | 4.659 | 4.7538 | -0.0948 | 5.1251 | 5.2361 | -0.111 | 5.2658 | 5.8433 | -0.5775 | 3.9275 | 4.0115 | -0.084 | 4.1348 | 4.4998 | -0.365 |
| FIG | 4.3368 | 4.2119 | 0.1249 | 4.5255 | 4.5143 | 0.0112 | 5.1004 | 5.0536 | 0.0468 | 5.5885 | 5.2709 | 0.3176 | 3.8839 | 3.8245 | 0.0594 | 4.3227 | 3.9944 | 0.3283 |
| BIG | 5.7597 | 4.0682 | 1.6915 | 5.8104 | 4.2271 | 1.5833 | 5.5494 | 4.1986 | 1.3508 | 5.8397 | 4.7798 | 1.0599 | 4.2773 | 3.8223 | 0.455 | 4.5892 | 4.0533 | 0.5359 |
| MFABA | 5.7783 | 4.0013 | 1.777 | 7.1895 | 4.0598 | 3.1297 | 5.6028 | 4.075 | 1.5278 | 6.2243 | 4.3752 | 1.8491 | 4.454 | 3.8081 | 0.6459 | 4.8555 | 3.9935 | 0.862 |
| AttEXplore | 6.4325 | 3.4516 | 2.9809 | 8.3455 | 3.4832 | 4.8623 | 6.2566 | 5.0651 | 1.1915 | 7.2563 | 5.2456 | 2.0107 | 6.0355 | 4.1755 | 1.86 | 6.9415 | 4.2988 | 2.6427 |
| GIG | 4.3707 | 4.4534 | -0.0827 | 4.6371 | 4.6904 | -0.0533 | 5.3644 | 5.4411 | -0.0767 | 5.4456 | 5.9125 | -0.4669 | 3.9515 | 3.9936 | -0.0421 | 4.4372 | 4.2935 | -0.2563 |
| EG | 4.9925 | 4.8459 | 0.1466 | 6.615 | 5.9209 | 0.6941 | 4.9839 | 4.6738 | 0.3101 | 5.6367 | 5.3364 | 0.3003 | 5.0065 | 4.906 | 0.1005 | 5.8683 | 5.6151 | 0.2532 |
| DeepLIFT | 4.1727 | 4.2879 | -0.1152 | 4.484 | 4.4823 | 0.0017 | 4.8818 | 5.1143 | -0.2325 | 5.1481 | 5.5994 | -0.4513 | 3.7093 | 4.0851 | -0.3758 | 3.8975 | 4.4789 | -0.5814 |
| SG | 4.173 | 4.1349 | 0.0381 | 4.2128 | 4.254 | -0.0412 | 6.2011 | 6.1561 | 0.045 | 6.1479 | 6.7031 | -0.5552 | 4.7479 | 4.7833 | -0.0354 | 4.8112 | 5.0538 | -0.2426 |
| AGI | 5.5176 | 4.0858 | 1.4318 | 8.2868 | 4.078 | 4.2088 | 5.8622 | 4.5894 | 1.2728 | 7.5437 | 4.431 | 3.1127 | 5.2245 | 4.1857 | 1.0388 | 7.6075 | 4.489 | 3.1185 |
| LAAI (our) | 7.6129 | 3.6638 | **3.9491** | 10.802 | 3.789 | **7.013** | 6.3754 | 3.9806 | **2.3948** | 7.6201 | 4.2561 | **3.364** | 6.0499 | 4.4146 | **1.6353** | 7.9069 | 4.6668 | **3.2401** |

## 3.5 LABEL-AGNOSTIC ATTRIBUTION FOR INTERPRETABILITY (LAAI)

Let $\{x^t\}_{t=0}^T$ denote the sequence of intermediate inputs along the adversarial path constructed by the update in Equation 4, and let $\Delta x^t = x^t - x^{t-1}$ be the perturbation at step $t$.

$$A(x_i) = \int \Delta x^t \cdot \frac{\partial \sum_{j=1}^C \log P_j(x^t)}{\partial x^t} \, dt \qquad (5)$$

Equation 5 represents the core formula of the LAAI algorithm. Our goal is to identify features that *maximally disrupt* the model's decision *without* assuming that the current prediction is correct. Instead of decreasing the confidence of a single target class, we increase the predictive uncertainty by pushing the output distribution $P(x^t)$ toward the maximum-entropy distribution $Q$ (uniform over classes). As derived in Appendix A, maximizing the entropy of $P(x^t)$ (equivalently, minimizing $\text{KL}(P(x^t), Q)$) yields a gradient proportional to $\frac{\partial}{\partial x^t} \sum_{j=1}^C \log P_j(x^t)$, which leads directly to Equation 5. Because the objective $\sum_j \log P_j(x^t)$ treats all classes symmetrically and does not depend on a label $y$, any feature whose perturbation either decreases the current class or increases competing classes will contribute to the attribution. This label-agnostic, uncertainty-driven objective directly mitigates *Information Ignorance* and *Extra Information* as defined in Section 3.2.

In practice, the updates $\Delta x^t$ follow the targeted adversarial attack strategy from AGI (Pan et al., 2021). We constrain each step by $\|\Delta x^t\|_\infty \leq \epsilon$, where $\epsilon$ is typically limited to one pixel and $T$ is the total number of iterations. Since the importance of each feature dimension is calculated independently, for the $i$-th dimension the feature importance can be expressed as $A(x_i) = \int \Delta x_i^t \cdot \frac{\partial \sum_{j=1}^C \log P_j(x^t)}{\partial x_i^t} \, dt$. The LAAI algorithm satisfies the Sensitivity axiom and Implementation Invariance axiom, with the proof in Appendix B. The implementation details are presented in Algorithm 2 in Appendix C.

**Remark 2.** *Any feature changes that increase the model's predictive uncertainty can be captured by Equation 5, since the attribution is explicitly tied to how perturbations move the output distribution toward maximum entropy.*

Although LAAI is class-agnostic and does not model class directionality, it is intended to complement rather than replace class-specific or contrastive attribution methods.

## 4 EXPERIMENTS

### 4.1 DATASET AND MODELS

Following previous work such as AGI (Pan et al., 2021), MFABA (Zhu et al., 2024), and AttEXplore (Zhu et al., 2023), we randomly selected 1000 images from the ImageNet (Deng et al., 2009) to maintain the consistency of the experiment. For our models, we chose three classic deep learning architectures: Inception-v3 (Szegedy et al., 2016), ResNet-50 (He et al., 2016), and VGG16 (Simonyan & Zisserman, 2014). For parameters details please refer to Appendix D. More ablation experiment results are provided in the Appendix F.7. To further verify the generalizability of the

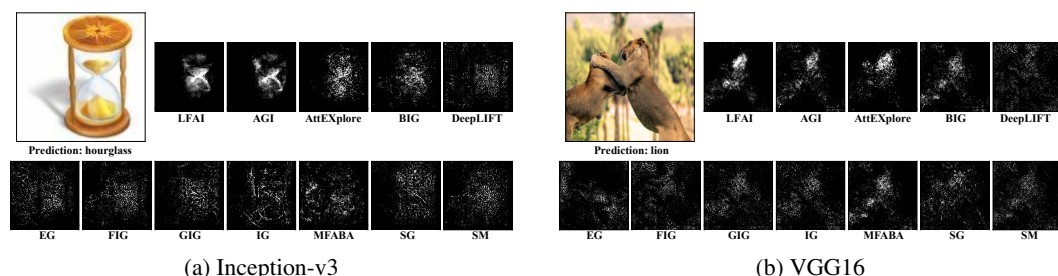

(a) Inception-v3                                    (b) VGG16

Figure 6: LAAI Attribution Results on Different Models (white: important, black: unimportant)

proposed LAAI method, we conducted additional experiments beyond the initial evaluation settings. Specifically, in the Appendix F.8 and F.9, we applied LAAI to the Vision Transformer ViT-B/16 (Dosovitskiy et al., 2020) model using the ImageNet dataset. Also to test dataset-level generalization, we evaluated LAAI on the VGG16 model using the CIFAR100 (Krizhevsky et al., 2009).

### 4.2 BASELINES

To comprehensively evaluate and compare our method, we selected 11 existing interpretability methods. The criteria for selection included publication in top-tier academic conferences and the availability of open-source code. The methods compared are AttEXplore (Zhu et al., 2023), AGI (Pan et al., 2021), MFABA (Zhu et al., 2024), BIG (Wang et al., 2021), EG (Erion et al., 2021), FIG (Hesse et al., 2021), DeepLIFT (Shrikumar et al., 2017), SG (Smilkov et al., 2017), SM (Simonyan et al., 2013), GIG (Kapishnikov et al., 2021), and IG (Sundararajan et al., 2017).

### 4.3 EVALUATION METRICS

We evaluate attribution methods using three metrics: (1) **Insertion/Deletion (INS/DEL)** (Samek et al., 2016) where INS inserts original pixels into a black baseline (calculating $\int_0^1 f(x^{(t)})dt$ with $x^{(0)}$ as all-black), while DEL replaces original pixels with black (same integral computation with $x^{(0)}$ as original); (2) **Fair Insertion/Deletion (F-INS/F-DEL)** using learned baseline $x^*$ where F-INS computes $\int_0^1 f(x^{(t)})dt$ by inserting original pixels into $x^*$, and F-DEL replaces original pixels with $x^*$; (3) **KL-INS/KL-DEL** measuring uncertainty dynamics through $\int_0^1 KL(P(x^*)\|P(x^{(t)}))dt$ (smaller KL-INS indicates faster uncertainty reduction) and $\int_0^1 KL(P(x^{\mathrm{orig}})\|P(x^{(t)}))dt$ (larger KL-DEL suggests faster uncertainty growth). We prioritize INS over DEL for consistency (Pan et al., 2021). More details are in Appendix E.

### 4.4 RESULTS

In this section, we provide additional experiments in Tables 1 and 2, where we further analyze the performance of various attribution methods on datasets split by confidence levels ($<70\%$ and $\geq 70\%$). More experiments, such as the performance on traditional attribution metrics like Insertion and Deletion, results on the full dataset without confidence level distinctions, results of F-INS and F-DEL, performance on Transformer-based models, and ablation studies can be found in Appendix F.

**INS and DEL:** In this section, we analyze the performance of our method LAAI compared to various state-of-the-art attribution methods using the Insertion Score (INS) and Deletion Score (DEL). Table 1 summarizes the performance of various interpretability methods, including LAAI, evaluated on Inception-v3, ResNet-50, and VGG16 models using the INS and DEL metrics. These classical INS/DEL scores are label-conditioned and depend only on changes in class confidence, without involving entropy or KL divergence, so they provide an evaluation criterion that is independent of our proposed KL-based metrics. Our method, LAAI, consistently outperforms other advanced attribution methods, achieving the highest GAP scores across all models. LAAI excels in both high-confidence and low-confidence datasets, providing superior interpretability by delivering more faithful explanations across different confidence levels. The average improvements of LAAI over other

methods are as high as 0.2232, 0.2049, and 0.1421 on the three models, respectively. Specifically, on high-confidence data, LAAI achieves an average GAP improvement of 0.2466 over other methods, demonstrating its superior performance under normal conditions. Additionally, LAAI achieves an average improvement of 0.099 on low-confidence data, indicating its robustness in addressing the Information Ignorance phenomenon. Moreover, we perform additional perturbation-level experiments, included in Appendix F.10, to illustrate that LAAI consistently provides optimal explanations across various degrees of input perturbations. These results further support the robustness of our approach under challenging conditions.

**KL-INS and KL-DEL:** In this section, we analyze the performance of our LAAI method and other interpretability methods using the KL Insertion (KL-INS) and KL Deletion (KL-DEL) metrics. These metrics evaluate the change in model decision uncertainty during the attribution process, providing a comprehensive assessment of how quickly the model's decision uncertainty is reduced or increased by the identified important features.

Table 2 shows the results across the three models, and the performance of our method, LAAI, demonstrates even more significant advantages compared to other attribution methods under this evaluation metric. Specifically, on low-confidence data, our LAAI method achieved an average GAP improvement of 2.1566, with respective improvements of 3.2499 on Inception-v3, 1.9132 on ResNet-50, and 1.3067 on VGG16. On high-confidence data, LAAI achieved an even more pronounced average GAP improvement of 3.7242, with respective gains of 5.7152 on Inception-v3, 2.7755 on ResNet-50, and 2.6820 on VGG16. These results highlight the substantial and consistent superiority of LAAI across both confidence levels.

**Attribution Results:** As shown in Figures 6a and 6b, the attribution results for both the VGG16 and Inception-v3 models demonstrate the superior performance of our LAAI method compared to other state-of-the-art attribution methods. LAAI consistently provides more precise and focused highlight regions on the critical features that contribute to the model's predictions. For instance, in the VGG16 model's prediction of a "lion," LAAI distinctly highlights the lion's mane and face, while other methods like AGI and AttEXplore show less distinct and more scattered focus areas. Similarly, for the Inception-v3 model's prediction of an "hourglass," LAAI accurately highlights the essential regions around the hourglass, unlike other methods that show broader and less precise attributions.

## 5 CONCLUSION

This paper identifies the phenomena of Information Ignorance and Extra Information, which can cause attribution bias in current gradient-based attribution algorithms. We propose a novel LAAI attribution algorithm that can avoid these issues and achieve accurate attributions. Additionally, we have designed a more comprehensive multi-dimensional attribution evaluation method. We also acknowledge that LAAI, like other perturbation-based attribution methods, does not explicitly constrain its adversarial trajectory to remain on the data manifold, and we highlight such "manifold-constrained paths" as an important future direction rather than an issue specific to LAAI. Looking towards future directions, our algorithm, like current gradient-based attribution algorithms, is primarily limited to visual models due to the continuity of pixels in image tasks. In future work, we will attempt to address the challenges of continuity in NLP tasks.

## ETHICS STATEMENT

We have read and will adhere to the ICLR Code of Ethics. This work uses only public data, involves no human subjects or personally identifiable information, and therefore does not require IRB review. Results are reported for research purposes only; we release anonymized code/configurations to support verification, and will disclose any funding sources and potential conflicts of interest upon acceptance.

## REPRODUCIBILITY STATEMENT

To support reproducibility, we release an anonymized repository with all experiment details including training/evaluation scripts, default hyperparameters, configuration files, and software/hardware environment.

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

## LLM USAGE DISCLOSURE

We used large language models (OpenAI GPT-4o and GTP-5) as auxiliary tools for grammar checking and language polishing of the manuscript. These models were not involved in research ideation, experimental design, implementation, or analysis. The authors take full responsibility for all content.

## A PROOF OF EQUATION 4

*Proof.*

$$\max - \sum_{j=1}^{C} P_j(x) \log P_j(x) \quad \text{s.t.} \quad \sum_{j=1}^{C} P_j(x) = 1$$

Construct the Lagrangian function: $\qquad(6)$

$$L\left(P_1(x), P_2(x), \ldots, P_C(x)\right) = \sum_{j=1}^{C} P_j(x) \log P_j(x) - \lambda \left( \sum_{j=1}^{C} P_j(x) - 1 \right)$$

$$\begin{cases} \frac{\partial L}{\partial P_j(x)} = - \log P_j(x) - 1 + \lambda = 0 & (1) \\ \sum_{j=1}^{C} P_j(x) - 1 = 0 & (2) \end{cases} \qquad(7)$$

substituting $P_j(x) = e^{\lambda - 1}$ into equation (1) $\qquad(8)$

$$P_j(x) = \frac{1}{C} \qquad(9)$$

We define the maximum entropy distribution $Q$, where the probability for class $j$ is $Q_j = \frac{1}{C}$. We aim to learn a model that maximizes the entropy for input $x$, which can be defined with the following loss function:

$$KL(P, Q) = \sum_{j=1}^{C} Q_j \log \frac{Q_j}{P_j(x)}$$

$$= \sum_{j=1}^{C} \frac{1}{C} \log^{\frac{1}{C}} - \frac{1}{C} \log P_j(x) \qquad(10)$$

$$\frac{\partial KL(P, \theta)}{\partial x} = -\frac{1}{C} \frac{\partial \sum_{j=1}^{C} \log P_j(x)}{\partial x} \qquad(11)$$

$$x - \alpha \cdot \text{sign}\left( \frac{\partial KL(P, \theta)}{\partial x} \right) = x + \alpha \cdot \text{sign}\left( \frac{\partial \sum_{j=1}^{C} \log P_j(x)}{\partial x} \right) \qquad(12)$$

$\square$

## B PROOF OF AXIOM

*Sensitivity Proof.* In the case where $\Delta x$ is sufficiently small, the first-order Taylor expansion holds. Given $|x| \leq \epsilon$, where $\epsilon$ represents a difference of 1 pixel value in this context, we have:

$$KL(Q \| P(x + \Delta x))$$

$$= \sum_{j=1}^{C} Q(x) \log P_j(x) + \Delta x \frac{\partial \sum_{j=1}^{C} Q(x) \log P_j(x)}{\partial x} \qquad(13)$$

$$= \frac{1}{C} \left[ \sum_{j=1}^{C} \log P_j(x) + \Delta x \frac{\partial \sum_{j=1}^{C} \log P_j(x)}{\partial x} \right]$$

Next, we observe the change in the KL divergence after updating $x$:

$$KL(Q\|P(x + \Delta x)) - KL(Q\|P(x))$$

$$= \frac{1}{C} \Delta x \frac{\partial \sum_{j=1}^{C} \log P_j(x)}{\partial x} \tag{14}$$

Accumulating all update terms, we obtain:

$$\int (KL(Q\|P(x^t + \Delta x)) - KL(Q\|P(x^t))) \, dt$$

$$= \int \frac{1}{C} \Delta x \frac{\partial \sum_{j=1}^{C} \log P_j(x)}{\partial x} \, dt$$

$$= KL(Q\|P(x^T)) - KL(Q\|P(x^0)) \tag{15}$$

$$\propto \underbrace{\int \Delta x \frac{\partial \sum_{j=1}^{C} \log P_j(x)}{\partial x} \, dt}_{LAAI}$$

Equation 15 demonstrates that when the distribution of the samples changes, a non-zero attribution is inevitably obtained, thus proving the sensitivity. $\square$

*Implementation Invariance Proof.* The Label-Agnostic Attribution for Interpretability (LAAI) algorithm adheres to the chain rule. Based on the properties of gradients, the LAAI algorithm satisfies implementation invariance, ensuring that results are consistent across different valid implementations of the same functional relationship. $\square$

## C   IMPLEMENTATION OF CONFUSION FEATURE ALGORITHM AND LABEL-AGNOSTIC ATTRIBUTION FOR INTERPREABILITY

---
**Algorithm 1** Confusion Feature Algorithm (CFA)

---
**Input:** Number of attack iterations $T$
**Output:** $x^{T+1}$
  1: $L = [\,]$
  2: $x^0 \sim U(0, 1)$
  3: **for** $t$ in $(1, ..., T+1)$ **do**
  4:     $x^t = x^{t-1} - \alpha \cdot \text{sign}\left( \frac{\partial \sum_{j=1}^{C} \log P_j(r(x^{t-1}))}{\partial x} \right)$
  5:     **if** convergence **then**
  6:        **break**
  7:     **end if**
  8:     Select $x^{T+1}$ from $L$ that maximizes $\mathcal{H}(x)$
  9: **end for**
  10: **Return:** $x^{T+1}$

---

## D   PARAMETERS

In all experiments, we used two NVIDIA A100 40GB GPUs. Our method involves two hyperparameters: the number of explorations $M$ and the number of attack iterations $T$, both set to 20.

## E   EVALUATION METRICS

We used four groups of evaluation metrics to assess the performance of the attribution methods.

The Insertion Score (INS) evaluates the area under the curve (AUC) as information from the original image is incrementally inserted into an all-black pixel image based on the attribution score, from high to low (Samek et al., 2016). This metric measures how quickly the current model decision

---

**Algorithm 2** Label-Agnostic Attribution for Interpretability (LAAI)

---

**Input:** Number of explorations $M$, number of attack iterations $T$
**Output:** $A$
1:   $A = 0$
2:   **for** $m$ in range $M$ **do**
3:     $x^0 = x$
4:     **for** $t$ in $(1, ..., T+1)$ **do**
5:       $x^t = x^{t-1} - \eta \cdot \text{sign}\left(\frac{\partial L(x^{t-1}, y_m)}{\partial x^{t-1}}\right)$
6:       $A \mathrel{+}= \eta \cdot \text{sign}\left(\frac{\partial L(x^{t-1})}{\partial x^{t-1}}\right) \cdot \frac{\partial \sum_{j=1}^{C} \log P_j(x^{t-1})}{\partial x^{t-1}}$
7:     **end for**
8:   **end for**
9:   **Return:** $A$

---

can be recovered. The Deletion (DEL) Score evaluates the AUC as pixels from the original image are progressively replaced with black pixels based on the attribution score, from high to low. This metric assesses how quickly the current model decision can be disrupted. **Compared to the DEL, the INS is generally considered more important** (Pan et al., 2021). Therefore, to eliminate the inconsistency between these two parameters in this paper, we use the GAP metric, which is the difference between INS and DEL, for unified comparison.

To address the issue of extra information in the evaluation process, we propose the Fair Insertion and Fair Deletion metrics. The Fair Insertion (F-INS) metric replaces the initial all-black image in the Insertion score with the learned $x^*$. Similarly, the Fair Deletion (F-DEL) metric replaces the zero-out operation in the Deletion score with the $x^*$ pixel operation. These modifications aim to provide a more accurate assessment by avoiding the introduction of extra information.

Furthermore, we introduce the KL Insertion (KL-INS) and KL Deletion (KL-DEL) metrics to evaluate the change in model decision uncertainty during the attribution process. KL Insertion replaces the current class output probability with the KL divergence $KL(Q, P(x))$ and calculates the AUC of the KL curve. A smaller area indicates that the important features identified by the attribution method quickly reduce the model's decision uncertainty. Conversely, KL Deletion calculates the AUC of the KL curve as important features are progressively removed, with a larger value indicating a rapid increase in model decision uncertainty. To illustrate this, we use the case where entropy is maximized. For example, in a scenario with 1000 classes, if the model's output probability for each class is exactly $\frac{1}{1000}$, the model's uncertainty reaches its maximum. This serves as a reference point for measuring the reduction in uncertainty during the attribution process.

# F   ADDITIONAL EXPERIMENTS

## F.1   RESULT OF INS AND DEL

Table 3 summarizes the evaluation results on three different models: Inception-v3, ResNet-50, and VGG16. Our method LAAI shows an average improvement of 0.1968 across all models. Specifically, LAAI achieved a GAP of 0.3410 on Inception-v3, 0.2752 on ResNet-50, and 0.1987 on VGG16. This represents average improvements of 0.2434, 0.2049, and 0.1421 over all other methods, and improvements of 0.1302, 0.1093, and 0.0768 over the three most advanced attribution methods (AGI, AttEXplore, and MFABA).

## F.2   RESULT OF F-INS AND F-DEL

In this section, we evaluate the performance of our LAAI method and other interpretability methods using the Fair Insertion (F-INS) and Fair Deletion (F-DEL) metrics. These metrics provide a fairer and more precise evaluation by mitigating the introduction of extra information.

Table 4 presents the performance results across the three models. Compared to other methods, LAAI demonstrates significant improvements, with an average increase of 0.1085 on low-confidence data.

Table 3: Evaluation of various interpretability methods via INS and DEL metrics. ↑ indicates that higher values in the column correspond to better interpretability performance, while ↓ indicates that lower values correspond to better interpretability performance. The * symbol denotes the primary reference metric for comparison.

| Method | Inception-v3 | | | ResNet-50 | | | VGG16 | | |
|---|---|---|---|---|---|---|---|---|---|
| | INS (↑) | DEL (↓) | GAP* (↑) | INS (↑) | DEL (↓) | GAP* (↑) | INS (↑) | DEL (↓) | GAP* (↑) |
| SM | 0.0701 | 0.0436 | 0.0265 | 0.0580 | 0.0323 | 0.0257 | 0.0426 | 0.0196 | 0.0230 |
| IG | 0.0787 | 0.0304 | 0.0483 | 0.0458 | 0.0223 | 0.0235 | 0.0312 | 0.0151 | 0.0162 |
| FIG | 0.0414 | 0.0587 | -0.0173 | 0.0284 | 0.0413 | -0.0129 | 0.0183 | 0.0286 | -0.0103 |
| BIG | 0.1577 | 0.0698 | 0.0879 | 0.1031 | 0.0609 | 0.0422 | 0.0688 | 0.0399 | 0.0289 |
| MFABA | 0.2281 | 0.0709 | 0.1572 | 0.1214 | 0.0681 | 0.0533 | 0.0865 | 0.0420 | 0.0444 |
| AttEXplore | 0.3066 | 0.0620 | 0.2446 | 0.2379 | 0.0466 | 0.1912 | 0.1870 | 0.0376 | 0.1494 |
| GIG | 0.1094 | 0.0414 | 0.0680 | 0.0522 | 0.0221 | 0.0302 | 0.0433 | 0.0182 | 0.0251 |
| EG | 0.3437 | 0.2890 | 0.0547 | 0.2763 | 0.2215 | 0.0548 | 0.2602 | 0.2126 | 0.0476 |
| DeepLIFT | 0.0944 | 0.0715 | 0.0229 | 0.0461 | 0.0358 | 0.0102 | 0.0413 | 0.0252 | 0.0161 |
| SG | 0.1887 | 0.0388 | 0.1499 | 0.1256 | 0.0246 | 0.1010 | 0.1300 | 0.0194 | 0.1106 |
| AGI | 0.2992 | 0.0684 | 0.2308 | 0.3103 | 0.0572 | 0.2531 | 0.2094 | 0.0374 | 0.1719 |
| LAAI (Ours) | 0.4276 | 0.0866 | **0.3410** | 0.3353 | 0.0601 | **0.2752** | 0.2383 | 0.0397 | **0.1987** |

Table 4: Evaluation of various interpretability methods via F-INS and F-DEL metrics.

| Method | Inception-v3 | | | | | | ResNet-50 | | | | | | VGG16 | | | | | |
|---|---|---|---|---|---|---|---|---|---|---|---|---|---|---|---|---|---|---|
| | Low Confidence (<70%) | | | High Confidence (≥70%) | | | Low Confidence (<70%) | | | High Confidece (≥70%) | | | Low Confidence (<70%) | | | High Confidence (≥70%) | | |
| | F-INS (↑) | F-DEL (↓) | GAP* (↑) | F-INS (↑) | F-DEL (↓) | GAP* (↑) | F-INS (↑) | F-DEL (↓) | GAP* (↑) | F-INS (↑) | F-DEL (↓) | GAP* (↑) | F-INS (↑) | F-DEL (↓) | GAP* (↑) | F-INS (↑) | F-DEL (↓) | GAP* (↑) |
| SM | 0.024 | 0.0207 | 0.0033 | 0.0549 | 0.0634 | -0.0085 | 0.0187 | 0.0294 | -0.0107 | 0.047 | 0.068 | -0.021 | 0.0133 | 0.0188 | -0.0055 | 0.0238 | 0.0408 | -0.017 |
| IG | 0.0306 | 0.0288 | 0.0018 | 0.0586 | 0.0934 | -0.0348 | 0.0222 | 0.0601 | -0.0379 | 0.0527 | 0.1305 | -0.0778 | 0.0172 | 0.0343 | -0.0171 | 0.0282 | 0.0788 | -0.0506 |
| FIG | 0.0276 | 0.0309 | -0.0033 | 0.0845 | 0.0746 | 0.0099 | 0.0561 | 0.0271 | 0.029 | 0.1188 | 0.0671 | 0.0517 | 0.0319 | 0.0205 | 0.0114 | 0.0729 | 0.033 | 0.0399 |
| BIG | 0.0802 | 0.0296 | 0.0506 | 0.1629 | 0.0648 | 0.0981 | 0.0488 | 0.035 | 0.0138 | 0.1399 | 0.0776 | 0.0623 | 0.0331 | 0.019 | 0.0141 | 0.0729 | 0.0411 | 0.0318 |
| MFABA | 0.0923 | 0.0288 | 0.0635 | 0.2632 | 0.0679 | 0.1953 | 0.0533 | 0.037 | 0.0163 | 0.1601 | 0.0877 | 0.0724 | 0.0377 | 0.0195 | 0.0182 | 0.1018 | 0.0422 | 0.0596 |
| AttEXplore | 0.1262 | 0.0218 | 0.1044 | 0.3503 | 0.0462 | 0.3041 | 0.0943 | 0.0279 | 0.0664 | 0.2881 | 0.0603 | 0.2278 | 0.0756 | 0.0161 | 0.0595 | 0.215 | 0.036 | 0.179 |
| GIG | 0.0288 | 0.0284 | 0.0004 | 0.0615 | 0.0889 | -0.0274 | 0.0198 | 0.0539 | -0.0341 | 0.0485 | 0.1163 | -0.0678 | 0.0154 | 0.0309 | -0.0155 | 0.027 | 0.0708 | -0.0438 |
| EG | 0.1306 | 0.1384 | -0.0078 | 0.3073 | 0.3241 | -0.0168 | 0.0838 | 0.0951 | -0.0113 | 0.2282 | 0.2769 | -0.0487 | 0.0629 | 0.0612 | 0.0017 | 0.1643 | 0.1433 | 0.021 |
| DeepLIFT | 0.0301 | 0.0296 | 0.0005 | 0.0779 | 0.0879 | -0.01 | 0.0273 | 0.0576 | -0.0303 | 0.0748 | 0.1182 | -0.0434 | 0.0157 | 0.0363 | -0.0206 | 0.031 | 0.0742 | -0.0432 |
| SG | 0.0241 | 0.0255 | -0.0014 | 0.0408 | 0.0779 | -0.0371 | 0.0124 | 0.0577 | -0.0453 | 0.0224 | 0.1156 | -0.0932 | 0.0096 | 0.0318 | -0.0222 | 0.0154 | 0.0781 | -0.0627 |
| AGI | 0.1211 | 0.0217 | 0.0994 | 0.3734 | 0.0555 | 0.3179 | 0.0808 | 0.026 | 0.0548 | 0.3784 | 0.0661 | 0.3123 | 0.0545 | 0.0157 | 0.0388 | 0.2538 | 0.039 | 0.2148 |
| LAAI (our) | 0.2139 | 0.0273 | **0.1866** | 0.5223 | 0.072 | **0.4503** | 0.1305 | 0.0368 | **0.0937** | 0.3881 | 0.0705 | **0.3176** | 0.0965 | **0.0164** | 0.0801 | 0.2738 | 0.0372 | **0.2366** |

Specifically, LAAI improves performance by 0.1583 on Inception-v3, 0.0927 on ResNet-50, and 0.0744 on VGG16. On high-confidence data, LAAI achieves an average GAP improvement of 0.2896, including a performance increase of 0.3784 on Inception-v3, 0.2835 on ResNet-50, and 0.2067 on VGG16. These results highlight the consistent and substantial advantages of LAAI over other methods in both low- and high-confidence scenarios, further establishing its robustness and effectiveness in interpretability tasks.

Table 5 presents the performance results across the three models. LAAI shows a pronounced improvement compared to other methods, with an average enhancement of 0.2333 over all methods. Specifically, LAAI demonstrates a performance gain of 0.3089 on Inception-v3, 0.2340 on ResNet-50, and 0.1569 on VGG16. Compared to the top three advanced methods (AGI, AttEXplore, and MFABA), LAAI shows an average improvement of 0.1082.

## F.3 RESULT OF KL-INS AND KL-DEL

Table 6 presents the results across the three models. For Inception-v3, LAAI achieved a GAP of 6.7771, indicating a significant reduction in model decision uncertainty with an average improvement of 5.5254 over all methods and 5.2004 over the top three advanced methods. On ResNet-50, LAAI achieved a GAP of 3.1944, with improvements of 2.6246 and 2.6196, respectively. For VGG16, LAAI attained a GAP of 2.8871, with enhancements of 2.3795 and 2.1526 over all methods and the top three advanced methods. These results underscore the robustness of our approach in enhancing model interpretability by effectively reducing decision uncertainty.

## F.4 PERFORMANCE ON TRANSFORMER-BASED MODEL VIT-B/16

In this section, we evaluate the performance of different attribution methods on the ViT-B/16 model, a widely-used transformer-based architecture in vision tasks. As shown in Table 8, our LAAI method outperforms others in both Insertion (INS) and Deletion (DEL) metrics, achieving the highest INS

Table 5: Evaluation of various interpretability methods via F-INS and F-DEL metrics.

| Method | Inception-v3 | | | ResNet-50 | | | VGG16 | | |
|---|---|---|---|---|---|---|---|---|---|
| | F-INS (↑) | F-DEL (↓) | GAP* (↑) | F-INS (↑) | F-DEL (↓) | GAP* (↑) | F-INS (↑) | F-DEL (↓) | GAP* (↑) |
| SM | 0.0461 | 0.0513 | -0.0052 | 0.0411 | 0.0599 | -0.0188 | 0.0201 | 0.0331 | -0.0130 |
| IG | 0.0507 | 0.0751 | -0.0244 | 0.0463 | 0.1158 | -0.0695 | 0.0244 | 0.0632 | -0.0388 |
| FIG | 0.0683 | 0.0622 | 0.0061 | 0.1057 | 0.0587 | 0.0470 | 0.0585 | 0.0286 | 0.0299 |
| BIG | 0.1394 | 0.0548 | 0.0846 | 0.1208 | 0.0687 | 0.0522 | 0.0589 | 0.0333 | 0.0256 |
| MFABA | 0.2147 | 0.0568 | 0.1579 | 0.1378 | 0.0771 | 0.0606 | 0.0793 | 0.0343 | 0.0451 |
| AttEXplore | 0.2867 | 0.0393 | 0.2474 | 0.2476 | 0.0535 | 0.1941 | 0.1661 | 0.0290 | 0.1371 |
| GIG | 0.0693 | 0.0846 | -0.0153 | 0.0501 | 0.1025 | -0.0524 | 0.0262 | 0.0497 | -0.0235 |
| EG | 0.3031 | 0.2508 | 0.0522 | 0.2379 | 0.1893 | 0.0486 | 0.1829 | 0.1589 | 0.0240 |
| DeepLIFT | 0.0844 | 0.0890 | -0.0046 | 0.0712 | 0.1030 | -0.0318 | 0.0277 | 0.0557 | -0.0280 |
| SG | 0.0561 | 0.0793 | -0.0232 | 0.0272 | 0.1106 | -0.0834 | 0.0147 | 0.0537 | -0.0390 |
| AGI | 0.3017 | 0.0459 | 0.2559 | 0.3162 | 0.0577 | 0.2586 | 0.1838 | 0.0309 | 0.1530 |
| LAAI (Ours) | 0.4347 | 0.0593 | **0.3754** | 0.3343 | 0.0635 | **0.2708** | 0.2116 | 0.0299 | **0.1817** |

Table 6: Evaluation of various interpretability methods via KL-INS and KL-DEL metrics.

| Method | Inception-v3 | | | ResNet-50 | | | VGG16 | | |
|---|---|---|---|---|---|---|---|---|---|
| | KL-INS (↑) | KL-DEL (↓) | GAP* (↑) | KL-INS (↑) | KL-DEL (↓) | GAP* (↑) | KL-INS (↑) | KL-DEL (↓) | GAP* (↑) |
| IG | 4.6330 | 4.7326 | -0.0996 | 5.2411 | 5.7370 | -0.4959 | 4.0892 | 4.3924 | -0.3032 |
| GIG | 4.6166 | 4.6722 | -0.0556 | 5.4314 | 5.8300 | -0.3986 | 4.0184 | 4.2275 | -0.2092 |
| SM | 4.2523 | 4.2874 | -0.0351 | 5.8697 | 5.9784 | -0.1087 | 4.1630 | 4.2865 | -0.1235 |
| SG | 4.2098 | 4.2448 | -0.0351 | 6.1572 | 6.6074 | -0.4501 | 4.7972 | 4.9943 | -0.1971 |
| DeepLIFT | 4.4600 | 4.4673 | -0.0073 | 5.1015 | 5.5145 | -0.4130 | 3.8561 | 4.3923 | -0.5362 |
| FIG | 4.5110 | 4.4910 | 0.0199 | 5.5031 | 5.2329 | 0.2702 | 4.2262 | 3.9570 | 0.2692 |
| EG | 6.4901 | 5.8381 | 0.6520 | 5.5225 | 5.2205 | 0.3020 | 5.6787 | 5.4591 | 0.2196 |
| BIG | 5.8065 | 4.2149 | 1.5916 | 5.7889 | 4.6781 | 1.1109 | 4.5206 | 4.0025 | 0.5181 |
| MFABA | 7.0808 | 4.0553 | 3.0255 | 6.1156 | 4.3226 | 1.7929 | 4.7672 | 3.9528 | 0.8145 |
| AGI | 8.0736 | 4.0786 | 3.9949 | 7.2495 | 4.4588 | 2.7907 | 7.0833 | 4.4223 | 2.6610 |
| AttEXplore | 8.1982 | 3.4808 | 4.7174 | 7.0813 | 5.2140 | 1.8674 | 6.7422 | 4.2717 | 2.4705 |
| LAAI (Ours) | 10.5564 | 3.7794 | **6.7771** | 7.4023 | 4.2079 | **3.1944** | 7.4984 | 4.6113 | **2.8871** |

score of 0.4357 and a low DEL score of 0.1067. This demonstrates that LAAI can effectively recover model decisions with minimal feature insertion while accurately identifying critical features whose removal significantly impacts the model. LAAI shows greater robustness compared to other methods, such as IG, FIG, and MFABA.

## F.5 RESULT OF 50% CONFIDENCE THRESHOLD EXPERIMENT RESULTS

High-confidence data indicates that the model is more confident in its decisions, suggesting fewer features interfere with the model's decision-making process. On the other hand, low-confidence data shows the model has less confidence in its decisions, implying that more features are affecting the model's decisions. This can be observed in the case shown in Figure 1. We also provide additional experiments in the Table 9 for the 50% confidence threshold. From the results, we can see that our LAAI method still performs the best.

## F.6 RESULT OF COMPUTATION EFFICIENCY

Table 10: Computation Efficiency of Different Methods

| | SM | IG | FIG | BIG | MFABA | AttEXplore | GIG | DeepLIFT | SG | EG | AGI | LAAI |
|---|---|---|---|---|---|---|---|---|---|---|---|---|
| FPS | 66.52 | 12.13 | 65.91 | 0.24 | 29.28 | 0.29 | 287.64 | 5.43 | 11.67 | 41.46 | 0.14 | 0.14 |

We tested the FPS of LAAI and various baseline methods. A higher FPS indicates faster speed. As shown, the computational cost of our LAAI method is comparable to that of AGI, but its performance is significantly improved over other methods. Therefore, we believe that this computational overhead is entirely acceptable.

Table 7: KL-INS and KL-DEL on High and Low Confidence Data

| | Inc-v3 | | | | Res-50 | | | | VGG16 | | | |
| | Low Conf (<70%) | | High Conf (≥70%) | | Low Conf (<70%) | | High Conf (≥70%) | | Low Conf (<70%) | | High Conf (≥70%) | |
| | KL-INS | KL-DEL | KL-INS | KL-DEL | KL-INS | KL-DEL | KL-INS | KL-DEL | KL-INS | KL-DEL | KL-INS | KL-DEL |
|---|---|---|---|---|---|---|---|---|---|---|---|---|
| SM | 4.0006 | 4.1461 | 4.2733 | 4.2992 | 5.7474 | 5.7749 | 5.8957 | 6.0216 | 4.0809 | 4.0889 | 4.1862 | 4.3423 |
| IG | 4.3213 | 4.478 | 4.659 | 4.7538 | 5.1251 | 5.2361 | 5.2658 | 5.8433 | 3.9275 | 4.0115 | 4.1348 | 4.4998 |
| FIG | 4.3368 | 4.2119 | 4.5255 | 4.5143 | 5.1004 | 5.0536 | 5.5885 | 5.2709 | 3.8839 | 3.8245 | 4.3227 | 3.9944 |
| BIG | 5.7597 | 4.0682 | 5.8104 | 4.2271 | 5.5494 | 4.1986 | 5.8397 | 4.7798 | 4.2773 | 3.8223 | 4.5892 | 4.0533 |
| MFABA | 5.7783 | 4.0013 | 7.1895 | 4.0598 | 5.6028 | 4.075 | 6.2243 | 4.3752 | 4.454 | 3.8081 | 4.8555 | 3.9935 |
| AttEXplore | 6.4325 | 3.4516 | 8.3455 | 3.4832 | 6.2566 | 5.0651 | 7.2563 | 5.2456 | 6.0355 | 4.1755 | 6.9415 | 4.2988 |
| GIG | 4.3707 | 4.4534 | 4.6371 | 4.6904 | 5.3644 | 5.4411 | 5.4456 | 5.9125 | 3.9515 | 3.9936 | 4.0372 | 4.2935 |
| EG | 4.9925 | 4.8459 | 6.615 | 5.9209 | 4.9839 | 4.6738 | 5.6367 | 5.3364 | 5.0065 | 4.906 | 5.8683 | 5.6151 |
| DeepLIFT | 4.1727 | 4.2879 | 4.484 | 4.4823 | 4.8818 | 5.1143 | 5.1481 | 5.5994 | 3.7093 | 4.0851 | 3.8975 | 4.4789 |
| SG | 4.173 | 4.1349 | 4.2128 | 4.254 | 6.2011 | 6.1561 | 6.1479 | 6.7031 | 4.7479 | 4.7833 | 4.8112 | 5.0538 |
| AGI | 5.5176 | 4.0858 | 8.2868 | 4.078 | 5.8622 | 4.5894 | 7.5437 | 4.431 | 5.2245 | 4.1857 | 7.6075 | 4.489 |
| LAAI (our) | 7.6129 | 3.6638 | 10.802 | 3.789 | 6.3754 | 3.9806 | 7.6201 | 4.2561 | 6.0499 | 4.4146 | 7.9069 | 4.6668 |

Table 8: Performance of different attribution methods on the ViT-B/16 model

| | IG | FIG | BIG | MFABA | AttEXplore | SM | GIG | EG | DeepLIFT | SG | AGI | LAAI (our) |
|---|---|---|---|---|---|---|---|---|---|---|---|---|
| INS | 0.1123 | 0.0616 | 0.225 | 0.2239 | 0.2749 | 0.1215 | 0.1052 | 0.285 | 0.0899 | 0.2082 | 0.3236 | 0.4357 |
| DEL | 0.0511 | 0.0968 | 0.1387 | 0.1724 | 0.1239 | 0.07 | 0.0461 | 0.269 | 0.0695 | 0.0299 | 0.1034 | 0.1067 |

## F.7 ABLATION EXPERIMENTS

Our method involves two key hyperparameters: the number of explorations ($M$) and the number of attack iterations ($T$). Both parameters were varied to observe their impact on different evaluation metrics, namely INS, DEL, GAP, F-INS, F-DEL, KL-INS, and KL-DEL. The values of $M$ and $T$ were set to 10, 15, 20, and 25 in our experiments. Detailed results of these ablation studies are provided in Table 11.

## F.8 PERFORMANCE OF DIFFERENT INTERPRETABILITY METHODS ON THE VIT-B/16 BACKBONE MODEL

## F.9 PERFORMANCE OF DIFFERENT INTERPRETABILITY METHODS ON THE CIFAR100 DATASET

## F.10 PERFORMANCE OF DIFFERENT INTERPRETABILITY METHODS UNDER INPUT PERTURBATIONS

Table 9: 50% Confidence Threshold Experiment Results

| | | SM | SG | MFABA | IG | GIG | FIG | EG | DeepLIFT | BIG | AttEXplore | AGI | LAAI |
|---|---|---|---|---|---|---|---|---|---|---|---|---|---|
| Conf<50% | INS | 0.0235 | 0.025 | 0.0876 | 0.0306 | 0.0284 | 0.0236 | 0.1206 | 0.028 | 0.0751 | 0.1193 | 0.1145 | 0.2104 |
| | DEL | 0.0179 | 0.0227 | 0.0264 | 0.0247 | 0.0251 | 0.0293 | 0.1324 | 0.0266 | 0.0275 | 0.0207 | 0.0198 | 0.0226 |
| Conf>=50% | INS | 0.0532 | 0.0396 | 0.2548 | 0.057 | 0.0597 | 0.0824 | 0.3002 | 0.0758 | 0.1597 | 0.3395 | 0.3609 | 0.5056 |
| | DEL | 0.0618 | 0.0757 | 0.0664 | 0.091 | 0.0865 | 0.0726 | 0.3153 | 0.0854 | 0.0635 | 0.0451 | 0.0541 | 0.0709 |

Table 11: Ablation study results for different hyperparameter settings of explorations ($M$) and attack iterations ($T$). The metrics include Insertion Score (INS), Deletion Score (DEL), Fair Insertion Score (F-INS), Fair Deletion Score (F-DEL), KL Insertion (KL-INS), and KL Deletion (KL-DEL). The GAP metric represents the difference between the respective insertion and deletion scores.

| $M$ | $T$ | INS | DEL | GAP | F-INS | F-DEL | GAP | KL-INS | KL-DEL | GAP |
|---|---|---|---|---|---|---|---|---|---|---|
| 10 | 10 | 0.5753 | 0.1070 | 0.4683 | 0.5713 | 0.0778 | 0.4936 | 10.2542 | 3.7087 | 6.5455 |
| | 15 | 0.5804 | 0.1092 | 0.4711 | 0.5768 | 0.0790 | 0.4978 | 10.3205 | 3.7195 | 6.6010 |
| | 20 | 0.5826 | 0.1097 | 0.4729 | 0.5815 | 0.0797 | 0.5018 | 10.3568 | 3.7395 | 6.6174 |
| | 25 | 0.5840 | 0.1109 | 0.4730 | 0.5837 | 0.0805 | 0.5032 | 10.3743 | 3.7447 | 6.6296 |
| 15 | 10 | 0.5851 | 0.1106 | 0.4745 | 0.5835 | 0.0803 | 0.5032 | 10.4150 | 3.7504 | 6.6646 |
| | 15 | 0.5887 | 0.1118 | 0.4769 | 0.5878 | 0.0827 | 0.5052 | 10.4560 | 3.7702 | 6.6858 |
| | 20 | 0.5907 | 0.1121 | 0.4786 | 0.5905 | 0.0827 | 0.5078 | 10.4815 | 3.7840 | 6.6976 |
| | 25 | 0.5913 | 0.1130 | 0.4782 | 0.5920 | 0.0840 | 0.5081 | 10.4902 | 3.7934 | 6.6968 |
| 20 | 10 | 0.5904 | 0.1121 | 0.4783 | 0.5903 | 0.0825 | 0.5078 | 10.4740 | 3.7864 | 6.6876 |
| | 15 | 0.5934 | 0.1132 | 0.4802 | 0.5929 | 0.0850 | 0.5079 | 10.5059 | 3.8044 | 6.7015 |
| | 20 | 0.5948 | 0.1135 | 0.4812 | 0.5946 | 0.0856 | 0.5089 | 10.5260 | 3.8149 | 6.7111 |
| | 25 | 0.5950 | 0.1144 | 0.4805 | 0.5963 | 0.0863 | 0.5100 | 10.5346 | 3.8238 | 6.7109 |
| 25 | 10 | 0.5936 | 0.1137 | 0.4799 | 0.5933 | 0.0844 | 0.5089 | 10.5115 | 3.8068 | 6.7047 |
| | 15 | 0.5961 | 0.1143 | 0.4818 | 0.5959 | 0.0861 | 0.5098 | 10.5397 | 3.8242 | 6.7155 |
| | 20 | 0.5963 | 0.1149 | 0.4814 | 0.5969 | 0.0867 | 0.5102 | 10.5520 | 3.8385 | 6.7134 |
| | 25 | 0.5967 | 0.1152 | 0.4816 | 0.5981 | 0.0875 | 0.5106 | 10.5621 | 3.8474 | 6.7148 |

Table 12: Interpretability performance on the ImageNet dataset (ViT-B/16).

| Method | Low Conf. Insertion | Low Conf. Deletion | High Conf. Insertion | High Conf. Deletion |
|---|---|---|---|---|
| SM | 0.0791 | 0.0330 | 0.1308 | 0.0782 |
| IG | 0.0603 | 0.0279 | 0.1238 | 0.0562 |
| FIG | 0.0294 | 0.0622 | 0.0687 | 0.1045 |
| BIG | 0.1354 | 0.0710 | 0.2452 | 0.1533 |
| MFABA | 0.1307 | 0.0765 | 0.2444 | 0.1937 |
| AttEXplore | 0.1241 | 0.0507 | 0.2269 | 0.1131 |
| GIG | 0.0582 | 0.0253 | 0.1145 | 0.0507 |
| EG | 0.1764 | 0.2069 | 0.3291 | 0.3909 |
| DeepLIFT | 0.0622 | 0.0284 | 0.0960 | 0.0786 |
| SG | 0.1475 | 0.0235 | 0.2210 | 0.0318 |
| AGI | 0.1044 | 0.0678 | 0.2234 | 0.1495 |
| **LAAI** | 0.1774 | 0.0531 | 0.3437 | 0.1269 |

Table 13: Interpretability performance on the CIFAR-100 dataset (VGG16).

| Method | Low Conf. Insertion | Low Conf. Deletion | High Conf. Insertion | High Conf. Deletion |
|---|---|---|---|---|
| SM | 0.0340 | 0.0409 | 0.0463 | 0.0391 |
| IG | 0.0325 | 0.0344 | 0.0428 | 0.0356 |
| FIG | 0.0367 | 0.0329 | 0.0367 | 0.0399 |
| BIG | 0.0427 | 0.0405 | 0.0577 | 0.0486 |
| MFABA | 0.0476 | 0.0408 | 0.0722 | 0.0494 |
| AttEXplore | 0.0592 | 0.0391 | 0.1157 | 0.0570 |
| GIG | 0.0323 | 0.0340 | 0.0421 | 0.0338 |
| EG | 0.1008 | 0.1099 | 0.1971 | 0.2257 |
| DeepLIFT | 0.0336 | 0.0327 | 0.0466 | 0.0350 |
| SG | 0.0403 | 0.0358 | 0.0694 | 0.0338 |
| AGI | 0.0443 | 0.0438 | 0.0655 | 0.0576 |
| LAAI | 0.1384 | 0.0438 | 0.2651 | 0.0661 |

Table 14: Interpretability performance across different perturbation-levels on the ImageNet dataset (Inception-v3). Here $\epsilon$ represents different perturbation rates.

| Method | $\epsilon$=2 | $\epsilon$=4 | $\epsilon$=6 | $\epsilon$=8 | $\epsilon$=10 | $\epsilon$=12 | $\epsilon$=14 | $\epsilon$=16 |
|---|---|---|---|---|---|---|---|---|
| SM | 0.0205 | 0.0178 | 0.0174 | 0.0224 | 0.0268 | 0.0285 | 0.0317 | 0.0208 |
| IG | 0.0138 | 0.0180 | 0.0234 | 0.0241 | 0.0292 | 0.0365 | 0.0428 | 0.0296 |
| FIG | 0.0193 | 0.0158 | 0.0231 | 0.0105 | 0.0193 | 0.0230 | 0.0114 | 0.0111 |
| BIG | 0.1147 | 0.1308 | 0.1093 | 0.1147 | 0.1208 | 0.1188 | 0.1228 | 0.0925 |
| MFABA | 0.1307 | 0.1443 | 0.1319 | 0.1229 | 0.1614 | 0.1596 | 0.1426 | 0.1379 |
| AttEXplore | 0.1311 | 0.1270 | 0.1242 | 0.1058 | 0.1331 | 0.1180 | 0.1312 | 0.1258 |
| GIG | 0.0150 | 0.0143 | 0.0200 | 0.0219 | 0.0222 | 0.0279 | 0.0255 | 0.0235 |
| EG | 0.1801 | 0.1784 | 0.1765 | 0.1775 | 0.1774 | 0.1796 | 0.1833 | 0.1790 |
| DeepLIFT | 0.0244 | 0.0317 | 0.0203 | 0.0291 | 0.0312 | 0.0211 | 0.0245 | 0.0260 |
| SG | 0.0518 | 0.0472 | 0.0542 | 0.0420 | 0.0544 | 0.0664 | 0.0509 | 0.0463 |
| AGI | 0.1801 | 0.1978 | 0.1890 | 0.1854 | 0.1778 | 0.1621 | 0.1617 | 0.1962 |
| LAAI | 0.2475 | 0.2464 | 0.2503 | 0.2411 | 0.2622 | 0.2614 | 0.2547 | 0.2654 |

