# LABEL-FREE ~~LABEL-FREE~~ LABEL-AGNOSTIC ATTRIBUTION FOR INTERPRETABILITY

## ABSTRACT

The importance of attribution algorithms in the AI field lies in enhancing model transparency, diagnosing and improving models, ensuring fairness, and increasing user understanding. Gradient-based attribution methods have become the most critical because of their high computational efficiency, continuity, wide applicability, and flexibility. However, current gradient-based attribution algorithms require the introduction of additional class information to interpret model decisions, which can lead to issues of information ignorance and extra information. Information ignorance can obscure important features relevant to the current model decision, while extra information introduces the incorrect identification of irrelevant features as significant. To address these issues, we propose the ~~Label-Free~~ Label-Agnostic Attribution for Interpretability (~~LFAI~~LAAI) algorithm, which analyzes model decisions without the need for specified class information. Additionally, to more rigorously assess the potential of current attribution algorithms, we introduce a variety of new evaluation metrics, combined with the traditional Insertion & Deletion Scores, to comprehensively assess the performance of our algorithm. To continuously advance research in the field of explainable AI (XAI), our algorithm is open-sourced at `https://anonymous.4open.science/r/LFAI-336C`.

## 1 INTRODUCTION

As deep learning advances, performance on tasks such as image recognition (Xu et al., 2023; Liu et al., 2023) has reached unprecedented levels, driving transformative applications in healthcare (Suganyadevi et al., 2022), autonomous driving (Grigorescu et al., 2020), and managerial decision-making (Shrestha et al., 2021). This growing reliance heightens the demand for transparent decisions: without explainability, users struggle to trust results or assign responsibility when failures occur.

Therefore, the research and development of Explainable AI (XAI) are of paramount importance. There has been extensive research in the XAI domain, with early interpretability methods such as Grad-CAM (Selvaraju et al., 2017) and LIME (Ribeiro et al., 2016) using heatmaps and local linear models to explain the decisions of Deep Neural Networks (DNNs). However, these methods have limitations in providing fine-grained and one-to-one explanations for each input feature. Consequently, researchers have proposed more detailed attribution methods, with Integrated Gradients (IG)~~Sundararajan et al. (2017)~~ (Sundararajan et al., 2017) being one of the most significant. IG addresses the shortcomings of earlier methods and introduces axioms for attribution, providing a consistent and fair framework for feature importance. As research progressed, new adversarial example-based attribution methods were proposed, such as Adversarial Gradient Integration (AGI)~~Pan et al. (2021)~~ (Pan et al., 2021), MFABA (Zhu et al., 2024), and AttEXplore (Zhu et al., 2023).

We note that existing attribution methods typically select specific class outputs or cross-entropy as the loss function and use backpropagation to obtain gradients concerning input samples to guide the attribution algorithm. We have identified two phenomena that cause attribution bias due to such gradient selection: **information ignorance and extra information.** Information ignorance refers to the omission of important features from classes not directly related to the model's final decision, while extra information involves the incorrect identification of irrelevant features as significant. The

former leads to interpretability methods overlooking many features crucial to the model's current decision and failing to explain low-confidence situations (applicable under any less-than-100% confidence conditions). And the latter results in feature leakage (Shah et al., 2021), where features not contributing to the model's decision are incorrectly identified as important.

~~We believe that attribution methods should not rely on labels (i.e., ground-truth class information or a specified class, such as the class with the highest confidence) . This is because a model'~~Instead of requiring label information as a hard prerequisite, we explore a complementary *label-agnostic* perspective on attribution. Models are indeed trained with labels, and label-conditioned explanations (e.g., "why is this image a dog?") remain important and widely used. However, many applications also demand answers to a different question: *which features drive the model's current predictive distribution?* without committing to any particular class. From this perspective, the model's output is fully determined by the input features, and ~~our goal is to explain why a specific output occurs and which features from the input are responsible for it. The output itself is independent of its label, and therefore, introducing labels during the explanation process is unnecessary~~the role of attribution is to characterize how these features shape the entire distribution rather than to justify a pre-selected label.

~~In fact, the inclusion of labels can exacerbate the problem of extra information, leading to the incorrect attribution of irrelevant features as significant.~~ When explanations are strictly tied to a single label—either the ground-truth or the top-1 prediction—the gradient $\nabla_x L(f(x), y)$ is forced to answer "why $y$?", which can systematically under-represent features that primarily affect competing classes (i.e., *information ignorance*) and over-emphasize features that merely support the hypothesis "this input is $y$" even when the predictive distribution is ambiguous (i.e., *extra information*). *To specify certain information always means to ignore what is not specified.* ~~Moreover, specifying a particular class can neglect the information from other classes , resulting in information ignorance, where important features for the decision may be overlooked.~~

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

$$\exists |\varphi| \geq k \quad \text{s.t.} \quad \varphi = \{i \mid i \in \Phi \text{ and } a_i < \tau\}$$

~~where $\varphi$ is the set of features with importance values $a_i$ below a threshold $\tau$, representing features that are not activated,~~

Formally, for a fixed classifier $f$, input $x \in \mathbb{R}^d$, and an attribution method $A$ that produces scores $a_i(x)$ for each feature $i \in \{1, \ldots, d\}$, we first fix an (oracle) relevant set $\Phi(x) \subseteq \{1, \ldots, d\}$ and thresholds $k \geq 1$ and ~~$\Phi$ denotes the true important regions. The parameter~~ $\tau > 0$. We then define

$$S_{\mathrm{II}}(x) := \{\, i \in \Phi(x) \mid a_i(x) < \tau \,\}. \tag{1}$$

We say that $A$ exhibits Information Ignorance at level $(k, \tau)$ on $x$ if $|S_{\mathrm{II}}(x)| \geq k$, i.e., at least $k$ ~~controls the amount of irrelevant features included. The role of $k$ is to allow the existence of some features that are actually important but are not assigned high attribution values during the attribution process~~ truly relevant features receive attribution scores below the threshold $\tau$.

**Extra Information** *(Informal)*: Extra Information refers to the inclusion of features from non-target classes that are not relevant for the decision process, which can mislead the attribution algorithm. In fact, the model initially inputs all features. Although the relative importance of these features varies, our aim is to discern which areas are crucial to the model's decisions and which are not. If certain unnecessary features are retained, they can blur the boundary between important and unimportant features. ~~Extra Information can be defined mathematically as:~~

$$\exists |\varphi| \geq k \quad \text{s.t.} \quad \varphi = \{i \mid i \notin \Phi \text{ and } a_i \geq \tau\}$$

Formally, we define

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

Equation 5 represents the core formula of the ~~LFAI algorithm. In the attribution process, we design the gradient to be accumulated as $\frac{\partial \sum_{j=1}^c \log P_j(x^t)}{\partial x^t}$, avoiding the phenomena of Information Ignorance and Extra Information.~~ LAAI algorithm. Our goal is to identify features that *maximally disrupt* the model's decision *without* assuming that the current prediction is correct. Instead of decreasing the confidence of a single target class, we increase the predictive uncertainty by pushing the output distribution $P(x^t)$ toward the maximum-entropy distribution $Q$ (uniform over classes). As derived in Appendix A, maximizing the entropy of $P(x^t)$ (equivalently, minimizing $\text{KL}(P(x^t), Q)$) yields a gradient proportional to $\frac{\partial}{\partial x^t}\sum_{j=1}^C \log P_j(x^t)$, which leads directly to Equation 5. Because the objective $\sum_j \log P_j(x^t)$ treats all classes symmetrically and does not depend on a label $y$, any feature whose perturbation either decreases the current class or increases competing classes will contribute to the attribution. This label-agnostic, uncertainty-driven objective directly mitigates *Information Ignorance* and *Extra Information* as defined in Section 3.2.

In practice, the updates $\Delta x^t$ ~~follows~~ follow the targeted adversarial attack ~~update~~ strategy from AGI (Pan et al., 2021). ~~Note that the only constraint on $\Delta x^t$ is $|\Delta x^t| \leq \epsilon$~~ We constrain each step by $\|\Delta x^t\|_\infty \leq \epsilon$, where $\epsilon$ is typically limited to one pixel and $T$ is the total number of iterations. Since the importance of each feature dimension is calculated independently, for the ~~i-th dimension,~~ $i$-th dimension the feature importance can be expressed as ~~$A(x_i) = \int \Delta x_i^t \cdot \frac{\partial \sum_{j=1}^c \log P_j(x_i^t)}{\partial x_i^t} dt$. Although the theoretical derivation is nontrivial, the resulting algorithm is elegant and easy to implement in practice. The LFAI~~ $A(x_i) = \int \Delta x_i^t \cdot \frac{\partial \sum_{j=1}^C \log P_j(x^t)}{\partial x_i^t} dt$. The LAAI algorithm satisfies the Sensitivity axiom and Implementation Invariance axiom, with the proof in Appendix B. The implementation details are presented in Algorithm 2 in Appendix C.

Table 2: Evaluation of various interpretability methods via KL-INS and KL-DEL metrics.

| | Inception-v3 | | | | | | ResNet-50 | | | | | | VGG16 | | | | | |
|---|---|---|---|---|---|---|---|---|---|---|---|---|---|---|---|---|---|---|
| | Low Confidence (<70%) | | | High Confidence (≥70%) | | | Low Confidence (<70%) | | | High Confidence (≥70%) | | | Low Confidence (<70%) | | | High Confidence (≥70%) | | |
| Method | KL-INS (↑) | KL-DEL (↓) | GAP* (↑) | KL-INS (↑) | KL-DEL (↓) | GAP* (↑) | KL-INS (↑) | KL-DEL (↓) | GAP* (↑) | KL-INS (↑) | KL-DEL (↓) | GAP* (↑) | KL-INS (↑) | KL-DEL (↓) | GAP* (↑) | KL-INS (↑) | KL-DEL (↓) | GAP* (↑) |
| SM | 4.0006 | 4.1461 | -0.1455 | 4.2733 | 4.2992 | -0.0259 | 5.7474 | 5.7749 | -0.0275 | 5.8957 | 6.0216 | -0.1259 | 4.0809 | 4.0889 | -0.008 | 4.1862 | 4.3423 | -0.1561 |
| IG | 4.3213 | 4.478 | -0.1567 | 4.659 | 4.7538 | -0.0948 | 5.1251 | 5.2361 | -0.111 | 5.2658 | 5.8433 | -0.5775 | 3.9275 | 4.0115 | -0.084 | 4.1348 | 4.4998 | -0.365 |
| FIG | 4.3368 | 4.2119 | 0.1249 | 4.5255 | 4.5143 | 0.0112 | 5.1004 | 5.0536 | 0.0468 | 5.5885 | 5.2709 | 0.3176 | 3.8839 | 3.8245 | 0.0594 | 4.3227 | 3.9944 | 0.3283 |
| BIG | 5.7597 | 4.0682 | 1.6915 | 5.8104 | 4.2271 | 1.5833 | 5.5494 | 4.1986 | 1.3508 | 5.8397 | 4.7798 | 1.0599 | 4.2773 | 3.8223 | 0.455 | 4.5892 | 4.0533 | 0.5359 |
| MFABA | 5.7783 | 4.0013 | 1.777 | 7.1895 | 4.0598 | 3.1297 | 5.6028 | 4.075 | 1.5278 | 6.2243 | 4.3752 | 1.8491 | 4.454 | 3.8081 | 0.6459 | 4.8555 | 3.9935 | 0.862 |
| AttEXplore | 6.4325 | 3.4516 | 2.9809 | 8.3455 | 3.4832 | 4.8623 | 6.2566 | 5.0651 | 1.1915 | 7.2563 | 5.2456 | 2.0107 | 6.0355 | 4.1755 | 1.86 | 6.9415 | 4.2988 | 2.6427 |
| GIG | 4.3707 | 4.4534 | -0.0827 | 4.6371 | 4.6904 | -0.0533 | 5.3644 | 5.4411 | -0.0767 | 5.4456 | 5.9125 | -0.4669 | 3.9515 | 3.9936 | -0.0421 | 4.4372 | 4.2935 | -0.2563 |
| EG | 4.9925 | 4.8459 | 0.1466 | 6.615 | 5.9209 | 0.6941 | 4.9839 | 4.6738 | 0.3101 | 5.6367 | 5.3364 | 0.3003 | 5.0065 | 4.906 | 0.1005 | 5.8683 | 5.6151 | 0.2532 |
| DeepLIFT | 4.1727 | 4.2879 | -0.1152 | 4.484 | 4.4823 | 0.0017 | 4.8818 | 5.1143 | -0.2325 | 5.1481 | 5.5994 | -0.4513 | 3.7093 | 4.0851 | -0.3758 | 3.8975 | 4.4789 | -0.5814 |
| SG | 4.173 | 4.1349 | 0.0381 | 4.2128 | 4.254 | -0.0412 | 6.2011 | 6.1561 | 0.045 | 6.1479 | 6.7031 | -0.5552 | 4.7479 | 4.7833 | -0.0354 | 4.8112 | 5.0538 | -0.2426 |
| AGI | 5.5176 | 4.0858 | 1.4318 | 8.2868 | 4.078 | 4.2088 | 5.8622 | 4.5894 | 1.2728 | 7.5437 | 4.431 | 3.1127 | 5.2245 | 4.1857 | 1.0388 | 7.6075 | 4.489 | 3.1185 |
| LAAI (our) | 7.6129 | 3.6638 | **3.9491** | 10.802 | 3.789 | **7.013** | 6.3754 | 3.9806 | **2.3948** | 7.6201 | 4.2561 | **3.364** | 6.0499 | 4.4146 | **1.6353** | 7.9069 | 4.6668 | **3.2401** |

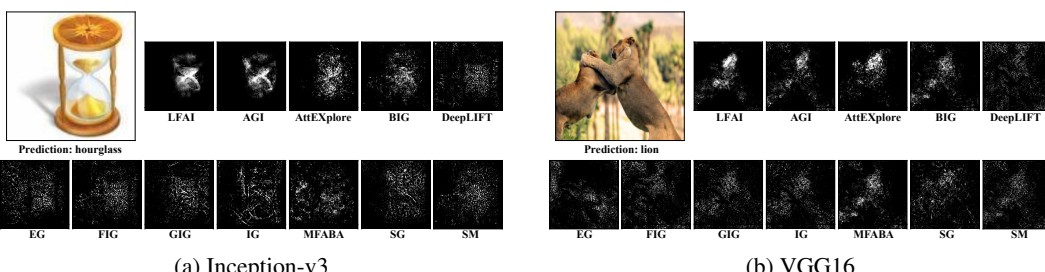

(a) Inception-v3    (b) VGG16

Figure 6: ~~LFAI~~ LAAI Attribution Results on Different Models (white: important, black: unimportant)

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

 INS (↑) | Inception-v3 DEL (↓) | Inception-v3 GAP* (↑) | ResNet-50 INS (↑) | ResNet-50 DEL (↓) | ResNet-50 GAP* (↑) | VGG16 INS (↑) | VGG16 DEL (↓) | VGG16 GAP* (↑) |
|---|---|---|---|---|---|---|---|---|---|
| SM | 0.0701 | 0.0436 | 0.0265 | 0.0580 | 0.0323 | 0.0257 | 0.0426 | 0.0196 | 0.0230 |
| IG | 0.0787 | 0.0304 | 0.0483 | 0.0458 | 0.0223 | 0.0235 | 0.0312 | 0.0151 | 0.0162 |
| FIG | 0.0414 | 0.0587 | -0.0173 | 0.0284 | 0.0413 | -0.0129 | 0.0183 | 0.0286 | -0.0103 |
| BIG | 0.1577 | 0.0698 | 0.0879 | 0.1031 | 0.0609 | 0.0422 | 0.0688 | 0.0399 | 0.0289 |
| MFABA | 0.2281 | 0.0709 | 0.1572 | 0.1214 | 0.0681 | 0.0533 | 0.0865 | 0.0420 | 0.0444 |
| AttEXplore | 0.3066 | 0.0620 | 0.2446 | 0.2379 | 0.0466 | 0.1912 | 0.1870 | 0.0376 | 0.1494 |
| GIG | 0.1094 | 0.0414 | 0.0680 | 0.0522 | 0.0221 | 0.0302 | 0.0433 | 0.0182 | 0.0251 |
| EG | 0.3437 | 0.2890 | 0.0547 | 0.2763 | 0.2215 | 0.0548 | 0.2602 | 0.2126 | 0.0476 |
| DeepLIFT | 0.0944 | 0.0715 | 0.0229 | 0.0461 | 0.0358 | 0.0102 | 0.0413 | 0.0252 | 0.0161 |
| SG | 0.1887 | 0.0388 | 0.1499 | 0.1256 | 0.0246 | 0.1010 | 0.1300 | 0.0194 | 0.1106 |
| AGI | 0.2992 | 0.0684 | 0.2308 | 0.3103 | 0.0572 | 0.2531 | 0.2094 | 0.0374 | 0.1719 |
| LAAI (Ours) | 0.4276 | 0.0866 | **0.3410** | 0.3353 | 0.0601 | **0.2752** | 0.2383 | 0.0397 | **0.1987** |

Table 4: Evaluation of various interpretability methods via F-INS and F-DEL metrics.

| Method | Inception-v3 Low Confidence (<70%) F-INS (↑) | Inception-v3 Low Confidence (<70%) F-DEL (↓) | Inception-v3 Low Confidence (<70%) GAP* (↑) | Inception-v3 High Confidence (≥70%) F-INS (↑) | Inception-v3 High Confidence (≥70%) F-DEL (↓) | Inception-v3 High Confidence (≥70%) GAP* (↑) | ResNet-50 Low Confidence (<70%) F-INS (↑) | ResNet-50 Low Confidence (<70%) F-DEL (↓) | ResNet-50 Low Confidence (<70%) GAP* (↑) | ResNet-50 High Confidece (≥70%) F-INS (↑) | ResNet-50 High Confidece (≥70%) F-DEL (↓) | ResNet-50 High Confidece (≥70%) GAP* (↑) | VGG16 Low Confidence (<70%) F-INS (↑) | VGG16 Low Confidence (<70%) F-DEL (↓) | VGG16 Low Confidence (<70%) GAP* (↑) | VGG16 High Confidence (≥70%) F-INS (↑) | VGG16 High Confidence (≥70%) F-DEL (↓) | VGG16 High Confidence (≥70%) GAP* (↑) |
|---|---|---|---|---|---|---|---|---|---|---|---|---|---|---|---|---|---|---|
| SM | 0.024 | 0.0207 | 0.0033 | 0.0549 | 0.0634 | -0.0085 | 0.0187 | 0.0294 | -0.0107 | 0.047 | 0.068 | -0.021 | 0.0133 | 0.0188 | -0.0055 | 0.0238 | 0.0408 | -0.017 |
| IG | 0.0306 | 0.0288 | 0.0018 | 0.0586 | 0.0934 | -0.0348 | 0.0222 | 0.0601 | -0.0379 | 0.0527 | 0.1305 | -0.0778 | 0.0172 | 0.0343 | -0.0171 | 0.0282 | 0.0788 | -0.0506 |
| FIG | 0.0276 | 0.0309 | -0.0033 | 0.0845 | 0.0746 | 0.0099 | 0.0561 | 0.0271 | 0.029 | 0.1188 | 0.0671 | 0.0517 | 0.0319 | 0.0205 | 0.0114 | 0.0729 | 0.033 | 0.0399 |
| BIG | 0.0802 | 0.0296 | 0.0506 | 0.1629 | 0.0648 | 0.0981 | 0.0488 | 0.035 | 0.0138 | 0.1399 | 0.0776 | 0.0623 | 0.0331 | 0.019 | 0.0141 | 0.0729 | 0.0411 | 0.0318 |
| MFABA | 0.0923 | 0.0288 | 0.0635 | 0.2632 | 0.0679 | 0.1953 | 0.0533 | 0.037 | 0.0163 | 0.1601 | 0.0877 | 0.0724 | 0.0377 | 0.0195 | 0.0182 | 0.1018 | 0.0422 | 0.0596 |
| AttEXplore | 0.1262 | 0.0218 | 0.1044 | 0.3503 | 0.0462 | 0.3041 | 0.0943 | 0.0279 | 0.0664 | 0.2881 | 0.0603 | 0.2278 | 0.0756 | 0.0161 | 0.0595 | 0.215 | 0.036 | 0.179 |
| GIG | 0.0288 | 0.0284 | 0.0004 | 0.0615 | 0.0889 | -0.0274 | 0.0198 | 0.0539 | -0.0341 | 0.0485 | 0.1163 | -0.0678 | 0.0154 | 0.0309 | -0.0155 | 0.027 | 0.0708 | -0.0438 |
| EG | 0.1306 | 0.1384 | -0.0078 | 0.3073 | 0.3241 | -0.0168 | 0.0838 | 0.0951 | -0.0113 | 0.2282 | 0.2769 | -0.0487 | 0.0629 | 0.0612 | 0.0017 | 0.1643 | 0.1433 | 0.021 |
| DeepLIFT | 0.0301 | 0.0296 | 0.0005 | 0.0779 | 0.0879 | -0.01 | 0.0273 | 0.0576 | -0.0303 | 0.0748 | 0.1182 | -0.0434 | 0.0157 | 0.0363 | -0.0206 | 0.031 | 0.0742 | -0.0432 |
| SG | 0.0241 | 0.0255 | -0.0014 | 0.0408 | 0.0779 | -0.0371 | 0.0124 | 0.0577 | -0.0453 | 0.0224 | 0.1156 | -0.0932 | 0.0096 | 0.0318 | -0.0222 | 0.0154 | 0.0781 | -0.0627 |
| AGI | 0.1211 | 0.0217 | 0.0994 | 0.3734 | 0.0555 | 0.3179 | 0.0808 | 0.026 | 0.0548 | 0.3784 | 0.0661 | 0.3123 | 0.0545 | 0.0157 | 0.0388 | 0.2538 | 0.039 | 0.2148 |
| LAAI (our) | 0.2139 | 0.0273 | **0.1866** | 0.5223 | 0.072 | **0.4503** | 0.1305 | 0.0368 | **0.0937** | 0.3881 | 0.0705 | **0.3176** | 0.0965 | **0.0164** | 0.0801 | 0.2738 | 0.0372 | **0.2366** |

confidence data. Specifically, ~~LFAI~~ LAAI improves performance by 0.1583 on Inception-v3, 0.0927 on ResNet-50, and 0.0744 on VGG16. On high-confidence data, ~~LFAI~~ LAAI achieves an average GAP improvement of 0.2896, including a performance increase of 0.3784 on Inception-v3, 0.2835 on ResNet-50, and 0.2067 on VGG16. These results highlight the consistent and substantial advantages of ~~LFAI~~ LAAI over other methods in both low- and high-confidence scenarios, further establishing its robustness and effectiveness in interpretability tasks.

Table 5 presents the performance results across the three models. ~~LFAI~~ LAAI shows a pronounced improvement compared to other methods, with an average enhancement of 0.2333 over all methods. Specifically, ~~LFAI~~ LAAI demonstrates a performance gain of 0.3089 on Inception-v3, 0.2340 on ResNet-50, and 0.1569 on VGG16. Compared to the top three advanced methods (AGI, AttEXplore, and MFABA), ~~LFAI~~ LAAI shows an average improvement of 0.1082.

## F.3 RESULT OF KL-INS AND KL-DEL

Table 6 presents the results across the three models. For Inception-v3, ~~LFAI~~ LAAI achieved a GAP of 6.7771, indicating a significant reduction in model decision uncertainty with an average improvement of 5.5254 over all methods and 5.2004 over the top three advanced methods. On ResNet-50, ~~LFAI~~ LAAI achieved a GAP of 3.1944, with improvements of 2.6246 and 2.6196, respectively. For VGG16, ~~LFAI~~ LAAI attained a GAP of 2.8871, with enhancements of 2.3795 and 2.1526 over all methods and the top three advanced methods. These results underscore the robustness of our approach in enhancing model interpretability by effectively reducing decision uncertainty.

## F.4 PERFORMANCE ON TRANSFORMER-BASED MODEL ViT-B/16

In this section, we evaluate the performance of different attribution methods on the ViT-B/16 model, a widely-used transformer-based architecture in vision tasks. As shown in Table 8, our ~~LFAI~~ LAAI method outperforms others in both Insertion (INS) and Deletion (DEL) metrics, achieving the high-

Table 5: Evaluation of various interpretability methods via F-INS and F-DEL metrics.

| Method | Inception-v3 | | | ResNet-50 | | | VGG16 | | |
|---|---|---|---|---|---|---|---|---|---|
| | F-INS ($\uparrow$) | F-DEL ($\downarrow$) | GAP* ($\uparrow$) | F-INS ($\uparrow$) | F-DEL ($\downarrow$) | GAP* ($\uparrow$) | F-INS ($\uparrow$) | F-DEL ($\downarrow$) | GAP* ($\uparrow$) |
| SM | 0.0461 | 0.0513 | -0.0052 | 0.0411 | 0.0599 | -0.0188 | 0.0201 | 0.0331 | -0.0130 |
| IG | 0.0507 | 0.0751 | -0.0244 | 0.0463 | 0.1158 | -0.0695 | 0.0244 | 0.0632 | -0.0388 |
| FIG | 0.0683 | 0.0622 | 0.0061 | 0.1057 | 0.0587 | 0.0470 | 0.0585 | 0.0286 | 0.0299 |
| BIG | 0.1394 | 0.0548 | 0.0846 | 0.1208 | 0.0687 | 0.0522 | 0.0589 | 0.0333 | 0.0256 |
| MFABA | 0.2147 | 0.0568 | 0.1579 | 0.1378 | 0.0771 | 0.0606 | 0.0793 | 0.0343 | 0.0451 |
| AttEXplore | 0.2867 | 0.0393 | 0.2474 | 0.2476 | 0.0535 | 0.1941 | 0.1661 | 0.0290 | 0.1371 |
| GIG | 0.0693 | 0.0846 | -0.0153 | 0.0501 | 0.1025 | -0.0524 | 0.0262 | 0.0497 | -0.0235 |
| EG | 0.3031 | 0.2508 | 0.0522 | 0.2379 | 0.1893 | 0.0486 | 0.1829 | 0.1589 | 0.0240 |
| DeepLIFT | 0.0844 | 0.0890 | -0.0046 | 0.0712 | 0.1030 | -0.0318 | 0.0277 | 0.0557 | -0.0280 |
| SG | 0.0561 | 0.0793 | -0.0232 | 0.0272 | 0.1106 | -0.0834 | 0.0147 | 0.0537 | -0.0390 |
| AGI | 0.3017 | 0.0459 | 0.2559 | 0.3162 | 0.0577 | 0.2586 | 0.1838 | 0.0309 | 0.1530 |
| LAAI (Ours) | 0.4347 | 0.0593 | **0.3754** | 0.3343 | 0.0635 | **0.2708** | 0.2116 | 0.0299 | **0.1817** |

Table 6: Evaluation of various interpretability methods via KL-INS and KL-DEL metrics.

| Method | Inception-v3 | | | ResNet-50 | | | VGG16 | | |
|---|---|---|---|---|---|---|---|---|---|
| | KL-INS ($\uparrow$) | KL-DEL ($\downarrow$) | GAP* ($\uparrow$) | KL-INS ($\uparrow$) | KL-DEL ($\downarrow$) | GAP* ($\uparrow$) | KL-INS ($\uparrow$) | KL-DEL ($\downarrow$) | GAP* ($\uparrow$) |
| IG | 4.6330 | 4.7326 | -0.0996 | 5.2411 | 5.7370 | -0.4959 | 4.0892 | 4.3924 | -0.3032 |
| GIG | 4.6166 | 4.6722 | -0.0556 | 5.4314 | 5.8300 | -0.3986 | 4.0184 | 4.2275 | -0.2092 |
| SM | 4.2523 | 4.2874 | -0.0351 | 5.8697 | 5.9784 | -0.1087 | 4.1630 | 4.2865 | -0.1235 |
| SG | 4.2098 | 4.2448 | -0.0351 | 6.1572 | 6.6074 | -0.4501 | 4.7972 | 4.9943 | -0.1971 |
| DeepLIFT | 4.4600 | 4.4673 | -0.0073 | 5.1015 | 5.5145 | -0.4130 | 3.8561 | 4.3923 | -0.5362 |
| FIG | 4.5110 | 4.4910 | 0.0199 | 5.5031 | 5.2329 | 0.2702 | 4.2262 | 3.9570 | 0.2692 |
| EG | 6.4901 | 5.8381 | 0.6520 | 5.5225 | 5.2205 | 0.3020 | 5.6787 | 5.4591 | 0.2196 |
| BIG | 5.8065 | 4.2149 | 1.5916 | 5.7889 | 4.6781 | 1.1109 | 4.5206 | 4.0025 | 0.5181 |
| MFABA | 7.0808 | 4.0553 | 3.0255 | 6.1156 | 4.3226 | 1.7929 | 4.7672 | 3.9528 | 0.8145 |
| AGI | 8.0736 | 4.0786 | 3.9949 | 7.2495 | 4.4588 | 2.7907 | 7.0833 | 4.4223 | 2.6610 |
| AttEXplore | 8.1982 | 3.4808 | 4.7174 | 7.0813 | 5.2140 | 1.8674 | 6.7422 | 4.2717 | 2.4705 |
| LAAI (Ours) | 10.5564 | 3.7794 | **6.7771** | 7.4023 | 4.2079 | **3.1944** | 7.4984 | 4.6113 | **2.8871** |

est INS score of 0.4357 and a low DEL score of 0.1067. This demonstrates that ~~LFAI~~ LAAI can effectively recover model decisions with minimal feature insertion while accurately identifying critical features whose removal significantly impacts the model. ~~LFAI~~ LAAI shows greater robustness compared to other methods, such as IG, FIG, and MFABA.

## F.5 RESULT OF 50% CONFIDENCE THRESHOLD EXPERIMENT RESULTS