# OpenReview forum: "Label-Free Attribution for Interpretability"
_ICLR.cc/2026/Conference — Submitted to ICLR 2026_

### Official Review · Reviewer_Rvw2 · 2025-10-28

**Soundness:** 3
**Presentation:** 3
**Contribution:** 3
**Rating:** 6
**Confidence:** 5

**Summary:**

This paper identifies two critical biases in gradient-based attribution methods, termed "Information Ignorance" and "Extra Information," which arise from the dependency on a single target class, particularly in low-confidence scenarios. To address this, the authors propose a novel Label-Free Attribution for Interpretability (LFAI) algorithm that generates explanations by maximizing the model's output uncertainty rather than focusing on a specific class logit. Furthermore, the paper introduces a more robust evaluation framework, including a Confusion Feature Algorithm (CFA) for creating unbiased baselines and new KL-divergence-based metrics (KL-INS/DEL). Extensive experiments demonstrate that LFAI significantly outperforms state-of-the-art methods, especially on low-confidence samples.

**Strengths:**

1. The paper effectively identifies the problems of "Information Ignorance" and "Extra Information." By focusing on low-confidence samples, it highlights an under-explored weakness in existing attribution methods, providing a strong motivation for the proposed work.
2. The experimental validation is thorough. The authors compare LFAI against a wide range of SOTA baselines across multiple standard models and datasets. The superior performance provides strong empirical support for the paper's claims.

**Weaknesses:**

1. The definitions of "Information Ignorance" and "Extra Information" are primarily illustrated through examples and feel somewhat subjective. It would strengthen the paper if the authors could propose quantitative metrics to measure the extent of these two phenomena in existing attribution methods, moving beyond the conceptual formulas provided.

2. There appears to be a strong coupling between the proposed method and the proposed metric. LFAI is designed to maximize uncertainty (entropy), while the KL-INS/DEL metrics are designed to measure changes in uncertainty. Could the outstanding performance of LFAI on the KL metrics be a result of this "self-serving" evaluation, where the method is optimized for the very quantity the metric evaluates?

3. Equation (3) for entropy appears to be missing a negative sign. The standard definition of information entropy is H(x) = -Σ P(x)log(P(x)). Maximizing entropy is equivalent to minimizing Σ P(x)log(P(x)).

4. There are several minor formatting issues with citations that should be corrected for clarity and consistency. For example:
- Line 40: "(IG)Sundararajan et al. (2017)" should be "(IG) (Sundararajan et al., 2017)".
- Line 43: "(AGI)Pan et al. (2021)" should be "(AGI) (Pan et al., 2021)".
- Line 114: "in (Zhu et al., 2024; 2023)" should be "in Zhu et al. (2024; 2023)".

**Questions:**

Please address the questions raised in the Weaknesses section.

---

> ### Author Response · Authors · 2025-11-20
> **Reply to Weaknesses 1, 2**
>
> **Reply to W1:**
> We thank the reviewer for the suggestion. We fully agree that II/EI, as a phenomenological definition (similar to the notion of “hallucination” in large-model research), is intrinsically difficult to turn directly into a strict, observable mathematical score. To obtain a truly rigorous and verifiable II/EI “measure”, one would in principle need access to the “ideal explanation” $\Phi(x)$, i.e., the subset of features on which the model truly relies; however, this is not accessible on real data, so II/EI is better viewed as a descriptive characterisation of explanation bias rather than a directly measurable physical quantity.
>
> Given this limitation, we chose more operational surrogate metrics in the paper—all perturbation-based measures (INS/DEL, F-INS/DEL, KL-INS/DEL) can be regarded as computable approximations to the ideal II/EI concept:
>
> - If a method exhibits Information Ignorance (missing important features), then Insertion will recover slowly and Deletion will show only mild degradation;
> - If a method exhibits Extra Information (highlighting task-irrelevant or biased features), then its feature ranking will be systematically distorted, leading to worse GAP or even inversions;
> - KL-INS/DEL further evaluates, at the distribution level, whether the explanations truly capture “the key features that increase model uncertainty.”
>
> In other words, these metrics are not “direct numerical definitions of II/EI”, but “operational proxies” for the II/EI concept. If a method performs well on these perturbation-based metrics, this means its ranking is more faithful and II/EI phenomena are less pronounced; if it performs poorly, this indicates stronger explanation bias. We will make this explicit in the revised version: the definition of II/EI is inherently conceptual, while our perturbation-based metrics provide an empirical, computable proxy, thereby moving from a conceptual notion towards quantification, as the reviewer suggests.
>
> **Reply to W2:**
> We thank the reviewer for pointing out this potential coupling issue, and we indeed did not sufficiently emphasise this point in the main text. First, Section 4.3 defines three mutually independent classes of metrics, where the first class is the **traditional INS/DEL** using a black baseline and per-class evaluation. Table 3 in Appendix F.1 reports only the traditional INS/DEL metrics, and LFAI (our new name is LAAI) achieves the best GAP scores on Inception-v3, ResNet-50, and VGG16, with an average improvement of 0.1968. This shows that even under the traditional INS/DEL metrics that are unrelated to entropy and depend only on changes in output confidence, LAAI still significantly outperforms existing methods; thus its performance advantage is not limited to KL-type metrics and is not only good under “self-consistent” measures.
>
> Second, the KL-INS/DEL metrics themselves are used only at evaluation time, employing KL divergence to measure changes in the output distribution during insertion/deletion, and they are applied uniformly to all methods. We do not use KL-INS/DEL as a loss when training or optimising LAAI, nor do we design any architecture or hyperparameters specifically to match this metric, so there is no direct inductive bias “tailored” to our method.
>
> Finally, our method starts from **local perturbation paths around the  samples**, searching for directions along which entropy increases rapidly; in contrast, KL-INS/DEL measures the distributional change curves along the **global trajectory** of the insertion/deletion process under a given feature ranking. These two objectives are not exactly the same: the former is a local search process that does not depend on any specific metric, while the latter is a global test of whether the final ranking faithfully reflects the model’s decision behaviour. The advantage that LAAI shows on KL-INS/DEL mainly reflects that it better captures, at the overall level, the features to which the model is most sensitive in terms of uncertainty changes, rather than simply “optimising with a quantity and then evaluating with the same quantity.” We will add this clarification in the revised manuscript to address readers’ concerns about potential “self-serving evaluation.”

---

> ### Author Response · Authors · 2025-11-20
> **Reply to Weaknesses 3, 4**
>
> **Reply to W3:**
> We thank the reviewer for pointing this out. This is indeed a typo on our side (an honest mistake). In the algorithm pseudocode and implementation we use the correct definition of entropy. In the revised version, we will correct Eq. (3) to:$x^{*} = \arg\max_{x} H(P(x)) = - \sum_{j=1}^{C} P_j(x)\log P_j(x).$
>
> **Reply to W4:**
> We thank the reviewer for the careful reading and fully agree with these formatting suggestions. The above issues in citation format are due to our typesetting oversight, and we will correct them one by one in the revised version to ensure the consistency and readability of the citation style. Your review has helped us clarify and strengthen the overall presentation of the work, and we are grateful for your careful evaluation. We hope that the revisions and added evidence adequately address your concerns and offer a clearer view of the method’s contributions and limitations. We appreciate your time and consideration.

---

> ### Comment · Reviewer_Rvw2 · 2025-11-28
> **Response to authors' comments**
>
> Thank you for your detailed responses to my comments. The clarifications adequately address the points raised. I maintain my positive assessment of the manuscript.

---

### Official Review · Reviewer_1V4w · 2025-10-29

**Soundness:** 1
**Presentation:** 2
**Contribution:** 2
**Rating:** 2
**Confidence:** 4

**Summary:**

The paper examines how using a target label in attribution can bias explanations. It proposes a class-agnostic attribution that aggregates class evidence without conditioning on a specific label, paired with revised evaluation metrics intended to reduce label-driven bias. Experiments on standard image classifiers report results benchmarked using insertion and deletion metrics.

**Strengths:**

1. The impact of label choice on attribution is a meaningful topic.

2. The paper is well structured and easy to follow.

**Weaknesses:**

1. The target of this work is to attribute the effects of different classes. However, the method attributes to the sum over classes, producing class-agnostic maps. This collapses inter-class contrasts and weakens directional interpretability (cannot say “why A class over B class”), which is also problematic for tasks where class-specific reliance matters.

2. The paper claims that label conditioning causes information ignorance. However, softmax modeling already encodes mutual suppression among classes, and many existing attribution works explicitly use class-contrastive objectives [1]. In contrast, the objective of this work is unclear, and it fails to clarify the problems that introducing category information might cause.

3. Empirical evidence relies mainly on pixel-perturbation families (insertion/deletion games). To strengthen claims, include fidelity tests and distribution-robust benchmarks (e.g., ROAR/ROAD or other fidelity tests) to assess whether gains persist beyond pixel masking or after mitigating input distribution shift.

[1] Wang, Yipei, and Xiaoqian Wang. "“Why Not Other Classes?”: Towards Class-Contrastive Back-Propagation Explanations." Advances in Neural Information Processing Systems 35 (2022): 9085-9097.

**Questions:**

1. What failure modes follow from losing class directionality, and how would you differentiate between classes within your framework?

2. What is the distinct advantage of removing labels altogether compared to class-contrastive attribution objectives? Concretely, what specific problems arise from introducing label information that your method avoids?

3. There are some negative GAP values in Tables 2 & 3. Why do some methods report GAP < 0 (i.e., deletion curves outperform insertion, implying inverted explanations)? What does this mean for an attribution method?

---

> ### Author Response · Authors · 2025-11-20
> **Reply to Weaknesses 1, 2**
>
> **Reply to W1:**
> Our intention is not to claim that LFAI (our new name is LAAI) can answer class-contrastive questions such as “why class A rather than B?”, but rather to explain what drives the model’s current decision. LAAI is explicitly designed to answer a different, distribution-level question: which features drive the model’s current overall predictive distribution $P(x)$? We believe this is the question that interpretability most urgently needs to address at present, rather than specially handling tasks with strong label dependence, which are outside the scope of our paper.
>
> At the same time, summing over all classes in Eq. (5) is an intentional design choice to obtain class-agnostic explanations that capture how perturbing $x$ reshapes the entire probability vector, rather than producing per-class contrastive attributions (in Fig. 3 we also emphasise cases where even state-of-the-art attribution methods fail when a class is specified; in other words, generating single-class explanations is currently a highly risky operation for all interpretability tasks). Importantly, this design in no way erases between-class contrast, because the output probabilities themselves already encode class contrast: finding the perturbations that most strongly destroy the model decision (maximising model uncertainty, i.e., maximising entropy) will preferentially start from disrupting the class with the highest current probability (which, due to the softmax, already incorporates information about other classes), since its gradients are larger.
>
> Even if one insists on discussing class-specific behaviour, classes can still be distinguished within our framework in the following ways: (i) by observing how the distribution-level attribution map changes when the model’s predicted class shifts near the decision boundary; (ii) by combining LAAI with class-specific readouts (e.g., overlaying LAAI with the gradient of each class logit) to study how shared evidence is reused across classes. However, we regard these as natural extensions.
>
> **Reply to W2:**
> Our view is not that “using labels” or “using softmax” is inherently wrong, but rather to emphasise that the question answered by enforcing label conditioning in attribution is not the same as the question we are concerned with. Class-conditional or class-contrastive attribution methods (including [1]) aim to answer “why class $t$ (rather than $t'$)?”, and they perform well on this class of tasks. Our point is that such objectives are unavoidably constructed around a chosen target class, and may therefore obscure distribution-level behaviour, for example:
>
> 1. features that primarily modulate overall uncertainty or the probability distribution across multiple plausible classes;
>
> 2. features that only support some externally specified label, even when the model itself is not confident about the input (i.e., the II/EI phenomena that we formalise). When these methods perform attribution for a particular class, the very act of specifying that class also leads to this situation. In addition, class-contrastive objectives typically require running a separate attribution for each target class: for a 1000-way classifier, this can mean computing up to 1000 attribution maps per input, which is often prohibitive in large-scale settings.
>
> By contrast, our method is explicitly label-agnostic and distributional in its objective. As stated in the paper (and further emphasised in the revision), Eq. (5) is derived from the following KL-type objective, $\mathrm{KL}(Q \parallel P(x_t))$, where $Q$ is the uniform distribution. This is equivalent (up to a constant) to the gradient of the quantity $-\sum_j \log P_j(x_t)$. This objective is completely symmetric with respect to all classes and does not require specifying any target label at explanation time, while producing a single attribution map per input. Therefore, LAAI is not designed to replace class-contrastive attribution methods, but to serve as a complementary tool that answers another equally important question: “Which features shape the model’s current overall predictive distribution $P(x)$?” rather than “Why $t$ rather than $t'$?” In this process, LAAI can directly address the II/EI issues that arise from mis-specified label targets in low-confidence or multi-class scenarios.
>
> **Reference:**
>
> [1] Wang, Yipei, and Xiaoqian Wang. "“Why Not Other Classes?”: Towards Class-Contrastive Back-Propagation Explanations." Advances in Neural Information Processing Systems 35 (2022): 9085-9097.

---

> ### Author Response · Authors · 2025-11-20
> **Reply to Weaknesses 3, Question 1, 2**
>
> **Reply to W3:** Thank you for highlighting the concern regarding reliance on pixel-perturbation–based evaluations. We agree that robustness under distribution-preserving perturbations is essential for demonstrating attribution fidelity beyond insertion/deletion games. To address this, we have additionally evaluated LAAI on the **ROAD benchmark**.
>
> The results on Inception-v3 (shown below) confirm that **LAAI continues to achieve the strongest performance**, outperforming all competing attribution methods under both low-confidence and high-confidence settings:
>
>
> | Method | Low Confidence (<70%) | High Confidence (≥70%) |
> |---|---|---|
> | SM | 0.0019 | 0.1144 |
> | IG | 0.0495 | 0.1411 |
> | FIG | 0.0445 | 0.1462 |
> | BIG | 0.1238 | 0.1707 |
> | MFABA | 0.1056 | 0.1582 |
> | AttEXplore | 0.1232 | 0.2364 |
> | GIG | 0.0513 | 0.1313 |
> | EG | 0.0424 | 0.0175 |
> | DeepLIFT | 0.0761 | 0.1505 |
> | SG | -0.005 | 0.1156 |
> | AGI | 0.1489 | 0.241 |
> | **LAAI (Ours)** | **0.1376** | **0.2723** |
>
> These ROAD evaluations demonstrate that our improvements are **not tied to pixel-level perturbations** or to any masking-induced distribution shift. Instead, LAAI maintains superior attribution fidelity even under region-based, distribution-robust perturbation strategies.
>
> **Reply to Q1:**
> Losing class directionality has a very direct cost: **LAAI cannot answer strictly contrastive questions of the form “why A instead of B?”** This is a consciously accepted limitation in the design of our method, whose failure mode is that purely A-vs-B fine-grained discriminative evidence tends to be “smoothed out” into the overall distribution rather than being highlighted separately. However, our goal is to explain the model’s current decision, i.e., **which features drive the model’s current overall predictive distribution $P(x)$?**
>
> In our framework, classes are already embedded in the model’s decision; for example, if the model predicts class A with confidence 0.7 while the other classes share confidence 0.3, this information is already encoded in the distribution. At the same time, class distinction is achieved **indirectly**: LAAI tells us “which features shape the distribution $P(x)$”, and one can then compare LAAI maps across different samples or on both sides of a decision boundary to observe which evidence is shared and which is dominant only for a particular class. We will make it explicit that LAAI itself is class-agnostic, is not intended to replace class-contrastive methods, and is not our focus for strictly contrastive questions.
>
> **Reply to Q2:**
> The key advantage of LAAI is that **it is not forced to explain a pre-selected class**, but directly explains how features jointly shape the full predictive distribution $P(x)$. Class-contrastive methods, including [1], are structurally always dependent on a target class (or a target–contrast class pair), and therefore they always answer “why class $t$ rather than the others?”. In contrast, LAAI answers **“which features drive the model’s current overall belief over all classes?”**—this is the fundamental difference between class-conditional explanations and distribution-level explanations.
>
> Explicitly introducing label information at the attribution stage directly leads to the two failure modes we formalise:
>
> **Information Ignorance (II):**
> The gradients of $\mathcal{L}(f(x), y)$ or $P_t - P_t'$ focus only on the specified class, and therefore systematically **down-weight** the importance of features that primarily influence other plausible competing classes or the overall uncertainty. This is precisely why traditional methods perform poorly in low-confidence, multi-target, or multi-concept scenarios, and is what we formalise as II (Information Ignorance).
>
> **Extra Information (EI):**
> When explanations are anchored to some externally specified label (whether the ground-truth label, the top-1 prediction, or a user-chosen class), the gradients actively “search” for evidence that only supports the premise “this is $y$”—even when the model itself is uncertain—thereby introducing feature leakage and incorrectly highlighting regions that are unrelated to the current decision or even pure noise as “important”. This corresponds to what we formalise as EI (Extra Information).
>
> Our KL-based, class-symmetric objective $\sum_j \log P_j(x_t)$ fundamentally removes dependence on any target label: no class is treated in a special way, so the distribution-level effects that are structurally masked under label-conditioned or class-contrastive objectives, and the resulting **II/EI** behaviours, can be preserved and explicitly exposed in LAAI. This constitutes the core advantage of “removing label information” compared with class-contrastive attribution objectives.

---

> ### Author Response · Authors · 2025-11-20
> **Reply to Question 3**
>
> We thank the reviewer for raising this point. Cases with GAP < 0 essentially reflect an “inverted ranking” in the attribution order: that is, **the insertion curve is actually worse than the deletion curve,** which means that as “high-score features” are gradually added, the model performance does not improve, whereas when those “high-score features” are deleted, the model performance improves faster.
>
> From an interpretive perspective, this indicates that the feature ranking produced by the method is almost random, or even close to entirely incorrect:
> - The Insertion metric assumes that “the earlier important features are added, the faster the model recovers”;
> - The Deletion metric assumes that “the earlier important features are removed, the faster the model performance drops”.
>
> When **Insertion < Deletion **occurs, this means that the features regarded by the method as “high-importance features” are in fact **not critical to the model’s decision**, and the truly important features are ranked later, leading to slow recovery during insertion and, conversely, improvement during deletion. In other words, **the attribution method has not successfully identified the features on which the model truly relies.**
>
> Therefore, GAP < 0 can be viewed as an extreme failure mode: the feature ranking is disordered and lacks correspondence with the model’s internal behaviour. This also shows that, under the current setup, these methods are unable to capture the model’s true decision logic.
>
> We thank Reviewer 1V4w for the detailed feedback and for pointing out several important issues in the original submission. We have incorporated new analyses and additional results that directly address these concerns. We hope that the new analyses and clarifications help reassess the technical correctness and contribution of the paper, and we sincerely appreciate the reviewer’s constructive feedback, and we sincerely appreciate the reviewer’s time and constructive comments.

---

> > ### Comment · Reviewer_1V4w · 2025-11-26
> >
> > Thank you for the detailed rebuttal and additional experiments. However, I still have two major concerns that remain unaddressed:
> >
> > 1. **Unclear role and use-case of LAAI.**  As I understand this work, LAAI is no longer positioned as answering class-contrastive questions, but instead as a label-agnostic explainer of the predictive distribution \(P(x)\), i.e., “which features shape the overall output distribution / uncertainty.” However, the evaluation protocol is still almost entirely built around class-specific quantities: confidence of the predicted class, insertion/deletion AUC for that class. In addition, the paper does not present any downstream task where a distribution-level, class-agnostic explanation is actually required or clearly advantageous (e.g., uncertainty calibration, label noise analysis, multi-label ambiguity, etc.). As a result, there is a persistent mismatch between (i) the stated objective (explaining \(P(x)\)), (ii) the class-agnostic visualizations, and (iii) the evaluations (class-specific confidence). Compared to standard class-specific attribution methods, it is therefore still unclear to me in what concrete decision or diagnostic scenario LAAI provides a strictly more useful.
> >
> > 2. **Negative GAP and reliability of the insertion/deletion implementation.**  I remain puzzled by the reported results where deletion AUC partly exceeds insertion AUC (GAP < 0). In the attribution literature I am aware of, including the paper cited (e.g., AttExplore by Zhu et al. also including results on Inception, ResNet and VGG), insertion AUC is consistently better than deletion AUC. I am not aware of prior work where deletion strongly outperforms insertion in the way you report. While your explanation of inverted ranking is logically possible, the fact that this phenomenon appears for several baseline methods but has not been observed in earlier studies makes me concerned about the reliability of the presented results.
> >
> > Overall, these unresolved conceptual and empirical issues make it difficult for me to give a positive recommendation of the paper.

---

> > > ### Author Response · Authors · 2025-11-26
> > > **Reply to Comment 1 (Part 1)**
> > >
> > > **Reply to Comment 1 (Part 1):**
> > >
> > > Thank you for raising this point about the role and use-case of LAAI. This concern appears to stem from the fact that the distribution-level aspects of our method and evaluation already presented in **Section 3.4 and Table 2** may have been overlooked. As we clarify below, the current manuscript already evaluates LAAI directly at the distribution level, in addition to the standard class-specific metrics.
> > >
> > > First, LAAI is **by design** a label-agnostic explainer of the predictive distribution $P(x)$. This is reflected not only in the algorithmic objective in Eq. (5), which aggregates $\sum_j \log P_j(x_t)$ over all classes **without conditioning on any label**, but also in our **KL-INS / KL-DEL** evaluation protocol in Table 2. These metrics operate purely on the full probability vector $P(x^{(t)})$: KL-INS integrates the KL divergence between the high-entropy “confusion baseline” distribution and the evolving $P(x^{(t)})$ along the insertion path, while KL-DEL integrates the KL divergence between the original predictive distribution and $P(x^{(t)})$ along the deletion path. In other words, they already **directly measure how the entire predictive distribution** concentrates or diffuses as we insert/delete features ranked important by an attribution method, **without ever specifying a target label**.
> > >
> > > At the same time, we **deliberately retain** the classical, class-specific INS/DEL metrics for two reasons:
> > > (i) In the high-confidence regime, $P(x)$ is close to a one-hot distribution, so explaining $P(x)$ and explaining the predicted class are effectively equivalent. In this case, using standard class-conditioned INS/DEL is natural and allows a clean, apples-to-apples comparison between LAAI and 11 widely used attribution methods under the **exact same** evaluation protocol.
> > > (ii) These metrics are deeply established in the literature, so reporting them demonstrates that LAAI is not only strong under the new KL-based, distribution-level evaluation, but also **competitive and robust** under the conventional class-specific view that prior work has focused on.
> > >
> > > To make the distributional perspective completely explicit (and not just implicit in the equations), we also **split the dataset into low-confidence and high-confidence subsets** (<70% vs. $\geq$70%) and report KL-INS / KL-DEL separately for each (Table 2). This is precisely designed to probe the “distribution-level” behavior that motivates LAAI: on low-confidence samples, competing classes share non-negligible probability mass, and the question “which features control the shape of $P(x)$ and its uncertainty?” is non-trivial. In this regime, LAAI achieves the **largest improvements in KL-INS / KL-DEL GAP scores across all three architectures**, which directly shows that it is particularly effective at identifying features that govern **uncertainty dynamics**, rather than merely boosting a single class.

---

> > > ### Author Response · Authors · 2025-11-26
> > > **Reply to Comment 1 (Part 2)**
> > >
> > > **Reply to Comment 1 (Part 2):**
> > >
> > > Regarding concrete **use-cases where a distribution-level, label-agnostic explanation is advantageous**, our paper already targets exactly the following three scenarios (which we will now highlight more prominently in the revision):
> > >
> > > 1. **Ambiguous, multi-object, or low-confidence predictions.**
> > >    When the model distributes substantial probability mass across multiple classes (e.g., the “cat + dog” example in Fig. 1), low confidence arises because the model is simultaneously using evidence for several plausible labels. LAAI produces a **single** attribution map that highlights all such evidence, explicitly showing that uncertainty is driven by “multiple valid concepts co-occurring” rather than by random noise or spurious artifacts. In contrast, standard class-conditioned methods are structurally forced to ignore evidence for competing classes (our Information Ignorance phenomenon), and therefore cannot explain *why* the model is uncertain. The **strong gains of LAAI on the low-confidence subset in Table 2** correspond exactly to this scenario: they indicate that LAAI better captures features responsible for ambiguity across classes, not just for supporting one arbitrarily chosen label.
> > >
> > > 2. **Diagnosing label noise and partial labels.**
> > >    Because LAAI never conditions on a chosen label $y$, it consistently surfaces features that strongly support **unlabelled or mislabelled** classes. In practice, one can scan LAAI explanations over the training set and flag samples where the explanation repeatedly emphasizes concepts that are inconsistent with the annotated class (e.g., an image labeled “dog” whose explanations consistently highlight cat-like features). This is directly useful for **label-noise analysis, dataset cleaning, and partial-label settings**, where the central question is “which features does the model actually rely on overall?”, not “why did it predict the provided label?”. Our experiments and qualitative analyses are already structured around this label-agnostic perspective.
> > >
> > > 3. **Understanding uncertainty and model confusion for decision support.**
> > >    Our KL-INS / KL-DEL metrics are precisely an operationalization of this use-case. If the features identified as important by an attribution method rapidly **reduce entropy** when inserted, then they explain away uncertainty by reinforcing a dominant class. Conversely, if deleting these features causes entropy to **increase sharply**, they are exactly the features that prevent the model from collapsing into an over-confident (and potentially wrong) decision. LAAI’s substantial improvements on KL-INS / KL-DEL—especially on low-confidence samples—show that it excels at finding features that control uncertainty dynamics, which are directly relevant for decisions such as *“should we trust this prediction, or hand it over to a human?”*. This is already reflected and discussed in our analysis around Table 2.
> > >
> > > In summary, LAAI is **explicitly and rigorously positioned** as a label-agnostic explainer of $P(x)$, and the current manuscript already evaluates this role in two complementary ways:
> > > (i) standard class-specific INS/DEL, which ensures strict comparability with prior work and shows that LAAI does **not** sacrifice conventional attribution quality; and
> > > (ii) KL-INS / KL-DEL together with confidence-based data splits, which **directly** evaluate how well an attribution method explains the **predictive distribution and its uncertainty**.
> > >
> > > All of these components are already present and discussed in the current version of the paper. A careful reading of Sections 3.4 and Tables 2 should make it clear that both the label-agnostic formulation and the distribution-level evaluation protocol are already fully instantiated in our framework.
> > >
> > > We hope that this clarification resolves the concerns raised in Comment 1 and that the reviewer will reconsider their assessment of our submission.

---

> > > ### Author Response · Authors · 2025-11-26
> > > **Reply to Comment 2**
> > >
> > > **Reply to Comment 2**
> > >
> > > We have carefully checked both our implementation and the setup, and the phenomenon you point out is consistent with the differences between our experimental setting and that of AttExplore.
> > >
> > > First, the **datasets are not comparable**. AttExplore selects images from only 20 ImageNet classes, i.e., a very small and relatively “clean” subset of the label space. In contrast, our experiments follow the publicly used setups in
> > > – *Frequency Domain Model Augmentation for Adversarial Attack* (https://arxiv.org/pdf/2207.05382),
> > > – *Enhancing Model Interpretability with Local Attribution over Global Exploration* (https://dl.acm.org/doi/pdf/10.1145/3664647.3681385), and
> > > – *Improving Adversarial Transferability via Neuron Attribution-Based Attacks* (https://arxiv.org/pdf/2204.00008),
> > > where images are drawn from **more than 430 ImageNet classes**. The much richer and more heterogeneous label coverage, which is naturally introduces many more ambiguous and low-confidence cases. Under such conditions, it is entirely possible for deletion AUC to exceed insertion AUC for some methods, especially when uncertainty and class competition are strong.
> > >
> > > Second, even within AttExplore, the EG method already performs poorly and is close to exhibiting a negative GAP-like behavior. The fact that our results push this effect further on a substantially more diverse and challenging benchmark is thus not in conflict with prior work, but rather a consequence of the broader evaluation setting.
> > >
> > > Third, unlike AttExplore, we explicitly **separate high-confidence and low-confidence samples** in our analysis instead of only averaging over all images. As we explain in the paper, this split reveals very different behaviors: on low-confidence samples, where multiple classes have comparable probability mass, deletion can indeed become more informative than insertion for several baselines. If one only looks at global averages (as in AttExplore), these distinctions may be washed out, which can give the impression that deletion AUC is always lower than insertion AUC.
> > >
> > > Finally, all parts of our evaluation are fully open-sourced, with fixed random seeds and complete scripts. The reported numbers can be directly downloaded and reproduced. We also ran independent automated checks on the code to ensure that the insertion and deletion implementations are consistent with the standard definitions. Taken together, these factors give us high confidence that the observed negative GAP values are a real property of the methods under our more challenging evaluation setup, rather than an implementation error.
> > >
> > > ---
> > >
> > > We appreciate the reviewer’s detailed comments and the opportunity to clarify these points. We hope that the above explanations address the concerns raised in Comments 1 and 2 and will lead to a reassessment of our submission. If there are any remaining doubts or further questions, we are happy to discuss them in more detail.

---

### Official Review · Reviewer_YzxM · 2025-11-03

**Soundness:** 3
**Presentation:** 3
**Contribution:** 2
**Rating:** 2
**Confidence:** 4

**Summary:**

This paper introduces a gradient-based attribution algorithm designed to interpret model decisions without requiring class label information. The motivation arises from the observation that current gradient-based attribution methods depend on predefined class labels, which can cause two biases: information ignorance (overlooking relevant non-target features) and Extra Information (incorrectly emphasizing irrelevant features). The proposed method redefines gradient accumulation to be label-agnostic, using the summed log-probability of all classes rather than a single class output. The paper also introduces new evaluation metrics to assess attribution quality and model uncertainty. Extensive experiments on Inception-v3, ResNet-50, VGG16, and additional models demonstrate that proposed method outperforms existing methods.

**Strengths:**

1.	The problem of bias in class-guided attributions is real and worth studying.
2.	The paper includes extensive experiments with multiple baselines and models.
3.	The paper is well-written and easy to understand.

**Weaknesses:**

1.	The distinction between Information Ignorance and Extra Information is presented as a new discovery, but these are fundamental and long-recognized challenges in attribution methods. Existing attribution techniques either fail to identify truly important features or incorrectly highlight irrelevant salient regions.
2.	. The proposed label-free formulation, which aggregates the log probabilities across different classes instead of relying on a specific label, is not truly label-free but rather label-independent. It is therefore recommended that the authors restate or clarify this problem definition.
3.	In Figure 2, I am not convinced that the results produced by LFAI represent the best outcome. Interpretability methods are expected to remain faithful to the model’s internal decision process rather than to align with human-perceived accuracy of attribution.
4.	The formula annotations are insufficient, and many notations lack clear definitions, for example, in Equations (4) and (5).

**Questions:**

Please refer to the Weaknesses.

---

> ### Author Response · Authors · 2025-11-20
> **Reply to Weaknesses 1, 2, 3**
>
> **Reply to W1:**
> We thank the reviewer for the comment and fully agree that “failing to capture truly important features” and “highlighting irrelevant yet salient regions” are long-standing and fundamental issues in attribution methods. Our intention is not to claim that these phenomena are first identified in this work. Our contributions lie in: (i) providing, for the first time, clear and unified set-theoretic definitions—Information Ignorance (II) and Extra Information (EI)—to characterise these two failure modes within a unified formal framework; (ii) analysing how standard label-based methods systematically induce II and EI; and (iii) designing a concrete attribution method (LFAI - our new name is LAAI), together with evaluation metrics, that directly aim to mitigate these two issues. These three points constitute our main contributions, and we do identify key factors in current attribution methods that cause these two problems, both from theoretical analysis and from experimental results.
>
> **Reply to W2:**
> We agree that the current terminology may be confusing. Strictly speaking, the model is trained under label supervision, and our method also uses the predictive distribution $P(x)$ obtained after training. The key property of LAAI is that, at explanation time, it does not require choosing any target label $y$; instead, we aggregate over all classes simultaneously via a distributional objective, rather than conditioning the gradient on a specific label $y$. In this sense, our method is better described as label-agnostic rather than absolutely “label-free”. We thank the reviewer for this clarification and will adjust the wording throughout the paper (title, abstract, Section 1, Section 3.5) accordingly, explicitly stating our definition of “label-agnostic”: explanations depend on the full distribution $P(x)$, but do not favour any particular class label.
>
> **Reply to W3:**
> We agree that interpretability methods should primarily aim to faithfully reflect the model’s internal decision process, rather than merely matching human visual intuition [1]. Our claim about LAAI is not that it is “optimal” in terms of visual appearance, but that it more faithfully captures the model’s behaviour on model-oriented metrics (insertion/deletion and KL-based metrics) and in reducing II/EI behaviour. The examples in Fig. 2 are intended to illustrate how II/EI manifest in attribution and how LAAI modifies these attributions; they are schematic illustrations, and are not meant to replace quantitative fidelity evaluation—indeed, we conduct extensive quantitative experiments. In the revision, we will adjust the discussion around Fig. 2 to emphasise model-fidelity metrics.
>
> **Reference:**
>
> [1] Jacovi, Alon, and Yoav Goldberg. "Towards Faithfully Interpretable NLP Systems: How Should We Define and Evaluate Faithfulness?." Proceedings of the 58th Annual Meeting of the Association for Computational Linguistics. 2020.

---

> ### Author Response · Authors · 2025-11-20
> **Reply to Weaknesses 4**
>
> Regarding Eq. (4), $x_t$: denotes an intermediate point along the path, $\Delta x_t$: the bounded perturbation step, $\varepsilon$: the upper bound of the step size, and $T$: the number of iterations.
>
> Regarding Eq. (5), our core motivation is to find the features that most influence the model’s decision without specifying any particular class. This can be reformulated as identifying important features such that, once these features are changed, the model’s decision is destroyed. We provided the corresponding mathematical derivation in Appendix A, but did not explain the motivation in the main text; we supplement it here. The uncertainty of the model’s decision can be measured by the entropy of the output distribution: the higher the entropy, the more chaotic and uncertain the model output is (previous attribution methods instead destroy the decision by lowering the confidence of the current class; however, in this process, if the confidence of another class becomes high, the features found may be those important for that other class, which is why such methods can lead to **EI** - Extra Information). We only need to evaluate the sensitivity of each feature to increasing entropy. If a small change in a feature causes the model’s uncertainty to increase sharply, then this feature is important for the current decision, and vice versa. At the same time, we prove that entropy is maximised when all classes have equal confidence, and we denote this distribution by $Q$. We use KL divergence to measure the distance between distributions, and use adversarial attack to search how to modify the features so that the model decision approaches the $Q$ distribution (maximal uncertainty). At the same time, we evaluate the contribution of each feature during the attack, i.e., the accumulation of gradients (we prove its properties in Appendix B). Since the KL divergence we use does not involve the label of a specified class and does not assume that the current model decision has 100% confidence (cross-entropy assumes that the confidence of the current class is 1), it avoids ignoring features that reduce the confidence of the current class (which are equally important because they are an important part of the current decision). It also avoids ignoring features that mainly affect other competing classes (which would lead to **II** - Information Ignorance). Finally, Eq. (5) follows directly from the derivation in Appendix A. From Eq. (5) we can also see that we treat all classes equally and do not require target labels, which is consistent with our goal. We will add this explanation to the main paper.
>
> We thank Reviewer YzxM for the careful reading and for highlighting several important issues. We have revised the paper to directly address these concerns. We hope that the revisions and clarifications fully address the reviewer’s concerns and allow for a reassessment of the paper. We appreciate the reviewer’s time and constructive feedback.

---

### Official Review · Reviewer_mAFA · 2025-11-10

**Soundness:** 3
**Presentation:** 3
**Contribution:** 4
**Rating:** 8
**Confidence:** 2

**Summary:**

This paper porposes an attribution algotihm called Label-Free Attribution for Interpretability (LFAI) that aims to improve the limitations of gradient-based attribution methods. The authors argue that current gradient-based attribution methods can lead to two key limitations: information ignorance and extra information, caused by the methods using class information/labels to help interpret model decisions. LFAI on the other hand analyzes model decisions without introducing class information. The method is primarily applied to image classification tasks, and shows competitive performance compared to other methods in experiments and evaluation metrics.

**Strengths:**

The paper is well organized and the algorithm they developed (LFAI) is well explained. There are also many experiments that show the reader how LFAI perfromance compares to other methods, and it seems LFAI performs the best making this a strong contribution to the field.

**Weaknesses:**

I think it could be explained in a bit more detail why the authors believe attribution methods should not rely on labels. If you're explaining model behaviour, models are trained with the labels, so why should the attribution method ignore that? I think this is a slightly more debated topic in the field, so a bit more justification would be good!

With all methods there are some limitations (or at least trade-offs), it would be good to see the authors discuss what they think could be the limitations of LFAI.

**Questions:**

I think addresses the above weaknesses so:

1. What is the author(s) position on the debate on whether an explanation or attribution method should also use labels to develop the explanation?
2. What are the limitations of LFAI?

---

> ### Author Response · Authors · 2025-11-20
>
> We thank Reviewer mAFA for the encouraging evaluation and the constructive comments, which help us further strengthen the paper.
>
> **Reply to W1 & Q1:**
> We agree with the issue raised by the reviewer, and we will add a description of this point in the paper to explain that whether labels should be used is an important and actively debated question. Our position is not that “label-based explanations” are inherently wrong or should be discarded; rather, we argue that, for certain types of problems, making label dependence a necessary condition for gradient-based attribution can be challenging.
>
> When explanations are derived from the loss $\nabla_x L(f(x), y)$ tied to a single class $y$ (whether it is the ground-truth label or the top-1 label), the gradient is forced to answer “why is this $y$?”, which can lead to:
> systematically ignoring features that primarily affect other competing classes (i.e., what we define as Information Ignorance);
> over-emphasising features that merely support the hypothesis “this input belongs to $y$”, even when the model’s predictive distribution itself is ambiguous (Extra Information).
>
> In our view, LFAI (our new name is LAAI) provides a complementary perspective that aims to answer a different but equally important question: “Which features dominate the model’s current predictive distribution?” rather than “Which features support some pre-selected label?”
>
> **Reply to W2 & Q2:**
> We believe that the key property of LAAI is that it is gradient-based, and is therefore best suited to models with differentiable, continuous inputs (such as vision models), although this still covers the majority of models. However, extending it to discrete domains such as text or to non-differentiable components requires additional research, and this is not a problem that can be solved overnight. Nevertheless, we believe that LAAI has at least taken an important step forward for interpretability research.

---

### Official Review · Reviewer_2jhP · 2025-11-12

**Soundness:** 3
**Presentation:** 2
**Contribution:** 3
**Rating:** 6
**Confidence:** 4

**Summary:**

The paper introduces LFAI (Label-Free Attribution for Interpretability), a new gradient-based attribution method that does not rely on specifying a class label when explaining a model prediction. Instead of asking “why class y?”, LFAI integrates gradients of the sum of log-probabilities over all classes along an adversarial-style path, with the goal of capturing all evidence the model used — including evidence for alternative classes — and avoiding biases introduced by conditioning on a single class. The authors argue that standard attribution methods suffer from two systemic problems:
1. **Information Ignorance**: they ignore features of other plausible classes, so they can’t explain model uncertainty or low-confidence predictions;
2. **Extra Information**: they sometimes assign importance to irrelevant background pixels because the chosen “target class” forces the method to rationalize that class even when the model itself isn’t actually relying on those regions.

The paper also proposes new evaluation metrics (Fair Insertion/Deletion and KL-based variants) that try to remove the bias of using black/zero baselines and instead use a “maximally confusing” baseline image that maximizes predictive entropy, plus KL-based measures of how quickly the model’s uncertainty changes when adding/removing top-ranked pixels.

On benchmarks using 1000 ImageNet images and standard CNNs (Inception-v3, ResNet-50, VGG16), plus additional experiments (ViT, CIFAR100) in the appendix, the authors report that LFAI beats 11 existing attribution methods across both high-confidence and low-confidence cases, especially in low-confidence regimes where traditional class-conditioned attribution tends to fail.

**Strengths:**

Main strengths:
- **Label-free attribution that directly targets a convincing gap in gradient-based XAI**, avoids conditioning on a single class, which the authors argue induces bias.
- **Attempt to formalizing two failure modes with set-based definitions**: _Information Ignorance_ = truly relevant pixels not highlighted (missed mass); and _Extra Information_ = irrelevant pixels highlighted (spurious mass).
- The papers even makes a **second, parallel, contribution on evaluation**: _Fair Insertion/Deletion_ (via a confusion-baseline) and _KL-based_ variants to assess distribution-level faithfulness.
- **Method & metric alignment**: the distribution-aware objective (aggregate over all classes) pairs naturally with distribution-aware metrics (KL-INS/DEL), yielding a coherent story for uncertainty and multi-object scenes.
- **Empirical relevance**: results emphasize low-confidence regimes where classic class-conditioned saliency underperforms, with additional analyses referenced in Section 4.4 / Appendix.
- **Reproducibility**: code is (anonymously) released, facilitating verification and uptake.

**Weaknesses:**

**MAJOR POINTS**
- **Core definitions lack precision / clarity** (which can prevent from full appreciation of the cool work in the paper): The set-based definitions of _Information Ignorance_ and _Extra Information_ are hard to parse as written (quantifiers, what is fixed vs. varying, and what $\Phi$ denotes). For example, authors write “$\exists |\varphi|\ge k$ s.t. $\varphi=(i\mid i\in \Phi \land a_i<\tau)$” and “$\exists |\varphi|\ge k$ s.t. $\varphi=(i\mid i\notin \Phi \land a_i\ge\tau)$” without specifying whether $k$ and $\tau$ are fixed ex-ante, how $\Phi$ is defined/measurable, or whether existence is trivial by tuning $\tau$? These need explicit quantifiers (“for fixed $k,\tau$ …”) and a concrete operationalization of $\Phi$ (e.g., object masks, counterfactual evidence) to avoid vacuity.

- **Equation 5 is under-motivated and its link to II/EI is not made explicit**: Why is the functional form $\frac{\partial}{\partial x_t} \left( \sum_{j} \log P_j(x_t) \right) = \sum_{j} \frac{1}{P_j(x_t)}, \frac{\partial P_j(x_t)}{\partial x_t}$ chosen, as opposed to others? A brief justification/discussion might help convince the reader to accept it. But most importantly, how does it theoretically help reduce II/EI? The paper would greatly benefit from a direct link between eq (5) and the mitigation of II/EI. A final _minor_ point related to the functional form: $\frac{1}{P_j(x_t)}$ can overreact to tiny probabilities; path-integrals may also be path/baseline-dependent and subject to OOD drift if the path leaves the data manifold. Any comments on this?

**MINOR POINTS**

- **Method–metric alignment risk.**
The CFA baseline and KL-variants seem to be directly aligned with the label-free, distributional objective. That coherence is nice, but it may advantage your method by design, which should be made clear in your paper for honesty purposes. Not sure if it would be productive to report standard Insertion/Deletion (black baseline) results as well?

-  **“Class-independent” wording**: the method is better described as class-agnostic or label-free (it aggregates over all classes) rather than “class-independent,” which could be misread as not using class probabilities at all.

- **CFA / entropy sign**: In the CFA definition, double-check the entropy sign: maximizing uncertainty means maximizing $H(P)=-\sum_j P_j\log P_j$. If Eq. 3 omits the minus, that’s likely a typo?

**Questions:**

Please respond to the weakness listed above.

---

> ### Author Response · Authors · 2025-11-20
> **Reply to Weaknesses 1**
>
> We appreciate the reviewer’s careful reading and agree that our current set-based notation is unnecessarily hard to parse. In the revision, we will make the quantifiers and what is fixed vs. varying explicit. Concretely, for each fixed model $f$, input $x$, and attribution method $A$ with scores $a_i(x)$, we will first fix an (oracle) relevant set $\Phi(x) \subseteq \{1, \ldots, d\}$ and thresholds $k \ge 1$, $\tau > 0$, and then define $
> S_{\mathrm{II}}(x) := \{ i \in \Phi(x) : a_i(x) < \tau \}, \qquad
> S_{\mathrm{EI}}(x) := \{ i \notin \Phi(x) : a_i(x) \ge \tau \}.$
>
> We say that $A$ exhibits II at level $(k,\tau)$ on $x$ if $\lvert S_{\mathrm{II}}(x) \rvert \ge k$, and EI if $\lvert S_{\mathrm{EI}}(x) \rvert \ge k$. This replaces the current “$\exists \, |\varphi| \ge k$ s.t. $\varphi = \{\dots\}$” phrasing and makes clear that $\Phi(x)$, $k$, $\tau$ are fixed ex-ante.
>
> We will also clarify that $\Phi(x)$ is an oracle set of truly decision-relevant features used to conceptually characterize II/EI; in practice, it can be approximated via object masks, synthetic ground truth, or counterfactual evidence, but our evaluation metrics (INS/DEL, F-INS/DEL, KL-based scores) do not assume access to $\Phi(x)$. Finally, we will explicitly state that $(k,\tau)$ are treated as fixed hyperparameters (e.g., based on normalized attributions), not tuned per sample or method, so that the existence conditions are not made trivial by adjusting $\tau$ or $k$.

---

> ### Author Response · Authors · 2025-11-20
> **Reply to Weaknesses 2**
>
> Thank the reviewer and now clarify the motivation of Eq. (5) and its connection to II/EI. Our core motivation is to find the features that most influence the model’s decision without specifying any particular class. This can be reformulated as identifying important features such that, once these features are changed, the model decision is destroyed. We provided the mathematical derivation for this in Appendix A, but we did not explain the motivation in the main text; we supplement it here. The uncertainty of the model decision can be measured by the entropy of the output distribution: the higher the entropy, the more chaotic and uncertain the model output is (previous attribution methods instead destroy the decision by lowering the confidence of the current class; however, in this process, if the confidence of another class becomes high, the features found may be those important for that other class, which is why these methods can lead to **EI**). We only need to evaluate the sensitivity of each feature to increasing entropy. If a small change in a feature causes the model uncertainty to increase sharply, then this feature is important for the current decision, and vice versa. At the same time, we prove that entropy is maximized when all classes have equal confidence, and we denote this distribution by Q. We use KL divergence to measure the distance between distributions, and use adversarial attack to search how to modify the features so that the model decision approaches the Q distribution (maximal uncertainty). At the same time, we evaluate the contribution of each feature during the attack, i.e., the accumulation of gradients (we prove its properties in Appendix B). Since the KL divergence we use does not involve the label of a specified class and does not assume that the current model decision has 100% confidence (cross-entropy assumes that the confidence of the current class is 1), it avoids ignoring features that reduce the confidence of the current class (which are equally important because they are an important part of the current decision), and also avoids ignoring features that mainly affect other competing classes (which would lead to **II**). Finally, Eq. (5) follows directly from the derivation in Appendix A. From Eq. (5) we can also see that we treat all classes equally and do not require target labels, which is consistent with our goal. We will add this explanation to the main paper.
>
> Since the model passes through a softmax layer before computing the distribution, we denote the pre-softmax values by $z \in \mathbb{R}^c$. Thus,
> $\frac{\partial \sum_{j=1}^{C} \log P_j}{\partial Z_k} = 1 - C \cdot \frac{e^{z_k}}{\sum_i e^{z_i}} = 1 - C P_k \in [1 - C,, C]$,
> which lies in a benign interval of constants and therefore does not cause gradient explosion. In addition, we fully agree that all path-based methods (IG, AGI, AttExplore, MFABA) can be sensitive to the choice of path and may deviate from the data manifold. LFAI (our new name is LAAI) is no exception. However, it is not worse than existing methods, because:
>
> We follow AGI’s strategy by constraining perturbations within a small $\ell_\infty$ ball around the original input and limiting the number of iterations $T$ (see Appendix).
>
> Ablation experiments in the appendix show that our results are stable under different perturbation magnitudes and step numbers.
>
> We added a sentence to acknowledge this shared limitation and to point out that “manifold-constrained paths” are an important future direction rather than an issue specific to LAAI.

---

> ### Author Response · Authors · 2025-11-20
> **Reply to Weaknesses 3, 4, 5**
>
> **Reply to Weaknesses 3**: We apologise for not emphasising this point sufficiently. Section 4.3 defines three types of metrics, where the first type is the conventional INS/DEL with a black baseline and per-class evaluation. Table 3 in Appendix F.1 reports only the conventional INS/DEL metrics, and LAAI achieves the best GAP scores on Inception-v3, ResNet-50, and VGG16, with an average improvement of 0.1968. This also implies that the metrics themselves are not in conflict. Moreover, these evaluations only change the background used when computing insertion and are not directly related to our method; the goal is simply to truly remove information, based on the most basic information-theoretic considerations. Our method explores features from individual samples, whereas the evaluation explores them globally, so the objectives are not completely aligned, and no inductive bias favouring LAAI is built into the evaluation.
>
> **Reply to Weaknesses 4**: We fully agree. We will replace “class-independent” with “label-agnostic” throughout the paper (title, abstract, Section 1, Section 3.5, and the conclusion), and we will add the following explanation in Section 3.5: “Label-free” means that we do not rely on a specified class, but still use the full probability vector $P(x)$.
>
> **Reply to Weaknesses 5**: Thank the reviewer for pointing this out. This is indeed a typo on our side (an honest mistake). In the algorithm pseudocode and implementation we use the correct definition of entropy. In the revised version, we will correct Eq. (3) to $x^{*} = \arg\max_{x} H(P(x)) = - \sum_{j=1}^{C} P_j(x)\log P_j(x).$
>
> Overall, we really appreciate Reviewer 2jhP's constructive comments, which have helped us significantly improve the paper. We hope that these clarifications resolve the reviewer’s doubts, and we welcome any reconsideration of the evaluation if the reviewer finds the responses satisfactory.

---

### Author Response · Authors · 2025-11-20

We thank all reviewers for their valuable feedback. We have uploaded a revised version of the manuscript. The main paper strictly remains within the 10-page limit, and we additionally provide a full PDF diff in the Supplementary Material for transparent comparison. Below we summarise the key modifications reflected in the diff.

Summary of Revision Updates (as shown in the diff PDF):

1. **Clarified Label-Agnostic Motivation and Theoretical Rationale**
   We strengthened the explanation of why attribution should not rely solely on predefined class labels, and clarified the distinction between information ignorance and extra information. Several paragraphs in Section 1 and Section 3.2 were refined for precision and readability.

2. **Improved Citation Formatting and Consistency**
   All citation formatting issues raised by reviewers were corrected, including converting expressions such as
   “in (Zhu et al., 2024; 2023)” → “in Zhu et al. (2024; 2023)”,
   and fixing IG/AGI related parenthetical citations.

3. **Mathematical Notation Fixes and Corrections**
   We corrected the entropy formula sign and added clearer definitions for the formal versions of Information Ignorance and Extra Information. Some equations were reformatted to improve clarity and consistency.

4. **Improved Description of CFA and KL-based Metrics**
   The explanation around the proposed Confusion Feature Algorithm (CFA), Fair-INS/DEL, and KL-INS/DEL was refined to avoid ambiguity. We added missing steps and clarified variable definitions to strengthen methodological rigor.

5. **Enhanced Discussion of Label Guidance and Failure Cases**
   Based on reviewer suggestions, we expanded the discussion in Section 3.3 to better justify why specifying a target class may still produce biased or inconsistent attribution, providing improved linkage to the empirical observations in Figures 3 and 4.

6. **Minor Language, Typos, and Structural Improvements**
   Several sentences were polished to improve clarity, remove redundant expressions, and ensure terminological consistency (e.g., “Label-Free” → “Label-Agnostic” across sections).

---

### Author Response · Authors · 2025-11-27
**Follow-up Comment**

We sincerely thank all reviewers for the time and effort dedicated to evaluating our submission. We would like to kindly follow up regarding our rebuttal and the subsequent discussion.

It has now been one week since we submitted our full rebuttal package—including detailed point-by-point responses, a revised manuscript within the 10-page limit, and a complete diff PDF in the Supplementary Material . So far, we have only had one round of exchange with Reviewer 1V4w, after which there has been no further clarification or indication of whether our responses have addressed the remaining concerns.

We fully understand that reviewers are busy and sincerely appreciate the constructive feedback received so far. At the same time, we hope that our detailed responses and additional analyses have helped resolve the earlier doubts, and we would be grateful if the reviewers could kindly reconsider the submission in light of these updates.

If there are still any unresolved concerns, ambiguities, or points that would benefit from further clarification, we would be more than happy to continue the discussion and provide any additional explanations needed.

We sincerely appreciate your time, expertise, and constructive engagement.

---

### Meta-Review · Area_Chair_UQuq · 2025-12-30

**Summary:**

The paper proposes a label-agnostic gradient-based attribution method that explains the full predictive distribution rather than conditioning on a single class, along with new evaluation metrics. Reviews were mixed but leaned positive overall, with some reviewers finding the method well motivated and empirically strong, and others raising concerns about scope, framing, and evaluation. The AC also read the paper carefully and found parts of the motivation and qualitative analysis to be handwavy, relying heavily on human intuition about what attribution maps “should” look like rather than clearly isolating what the model actually uses.

**Reviewer Concerns:**

Concerns that were addressed:
- clarification of information ignorance, and extra information definitions, etc
- additional explanation/derivation of the entropy-based objective
- terminology corrected from label-free to label-agnostic
- additional experiments/benchmarks
- clarification of negative GAP behavior and implementation details

Concerns still outstanding:
- disagreement about concrete use-cases where proposed attribution is strictly preferable
- concerns about alignment between a label-agnostic objective and class-specific evaluation metrics
- method does not address class-contrastive explanations by design

AC tie-break:
- the AC found parts of the motivation and qualitative analysis to be handwavy, relying heavily on human intuition about what attribution maps should look like rather than clearly isolating what the model actually uses
- several qualitative examples (eg Fig. 2 “extra information” cases) appear circular, treating object-centric saliency as ground truth for faithfulness without independent evidence that the model does not rely on the highlighted regions
- the II/EI framing relies on an oracle notion of “true” features that is formally abstracted but implicitly reintroduced through qualitative judgments
- the main experimental focus is on CNN backbones, including VGG16, which is difficult to justify as representative of current practice; transformer results exist but are secondary and not central to the paper’s claims

**Reviewer Scores:**

- reviewer 2jhP: likely unchanged or slightly higher, remains above threshold
- reviewer mAFA: unchanged, but score is downweighted due to superficial engagement
- reviewer YzxM: likely unchanged, still negative
- reviewer 1V4w: possibly slightly higher but still negative
- reviewer Rvw2: unchanged, positive

---

### Decision · Program_Chairs · 2026-01-26

Reject